# LexSign: Learning Sign Language from Lexical Descriptions

## Abstract

Sign languages are well-defined natural languages that convey meaning through both manual postures and non-manual expressions. While recent methods effectively transcribe sign language videos into compact textual tokens, they often overlook the intrinsic subunit-level structures of sign language. In this work, we explore leveraging the hierarchical structure within lexical descriptions to enhance fine-grained sign language understanding. Specifically, we first construct LexSign, a large-scale dataset comprising both manually curated and automatically generated lexical descriptions of signs. To guarantee the quality of generated descriptions, we build LexSign-Bench, a benchmark to comprehensively evaluate the sign language understanding capability of Multi-modal Large Language Models (MLLMs), and further propose a perceive-then-summarize pipeline that leverages large foundation models to generate high-quality lexical descriptions. Based on the constructed LexSign, we propose Hierarchical Action-Language Interaction (HALI) that conducts hierarchical alignment between lexical descriptions and sign language videos to obtain more distinguishable and generalizable visual representations. Experimental results on public datasets demonstrate that incorporating the collected lexical descriptions with the proposed HALI significantly improves performance across different sign language understanding tasks.

## 1 Introduction

Sign language serves as a primary medium of communication within the Deaf community, but is not largely known by hearing individuals. To help mitigate this communication barrier, vision-based sign language understanding (SLU) has emerged and developed rapidly (Camgoz et al., 2018; 2020; Chen et al., 2022a;b; Zuo et al., 2023; Wong et al., 2024; Jiao et al., 2024; Li et al., 2025c; Guo et al., 2025), aiming to enable automatic recognition and translation of sign language from video input into textual or symbolic representations in a non-intrusive manner. However, these methods often leverage either coarse-grained annotations with limited semantic details and generalizability (*e.g.*, gloss [1]), or highly detailed symbolic systems that demand with extensive expert efforts and are difficult for non-experts to learn and apply (*e.g.*, SignWriting (Sutton, 2010) and HamNoSys (Hanke, 2004)). These limitations emphasize the need for scalable, fine-grained annotations that can distinguish similar signs and recognize unseen ones, vital for both practical SLU applications and deeper understanding of non-verbal communication (Ong & Ranganath, 2005; Bragg et al., 2019).

As well-defined natural languages, sign languages follow explicit linguistic rules and frequently employ iconic symbolism to establish body–object and body–body mappings, commonly referred to as perceptual and pantomimic iconicities (Pyers & Senghas, 2020; Sehyr et al., 2021). As illustrated in Fig. 1, lexical descriptions from sign language dictionaries (Costello, 2008; China Association of the Deaf, 2003) provide explicit and detailed performance instructions for individual signs, facilitating the construction of such mappings and thereby enhancing generalization. Moreover, Fig. 1c illustrates that a complex gloss can be decomposed into a combination of finer-grained glosses (*e.g.*, Engineer ≈ Size+Person), and distinctions between semantically similar signs often lie in subtle subunit[2] details (*e.g.*, Engineer *vs.* Player). These intrinsic properties of sign language underscore

---

[1]Gloss is a written approximation of a sign, typically reflecting its semantic meaning.

[2]Sign language subunit is the smallest component that can distinguish different sign, typically containing five terms: handshape, palm orientation, hand location, hand movement, and non-manual signal.

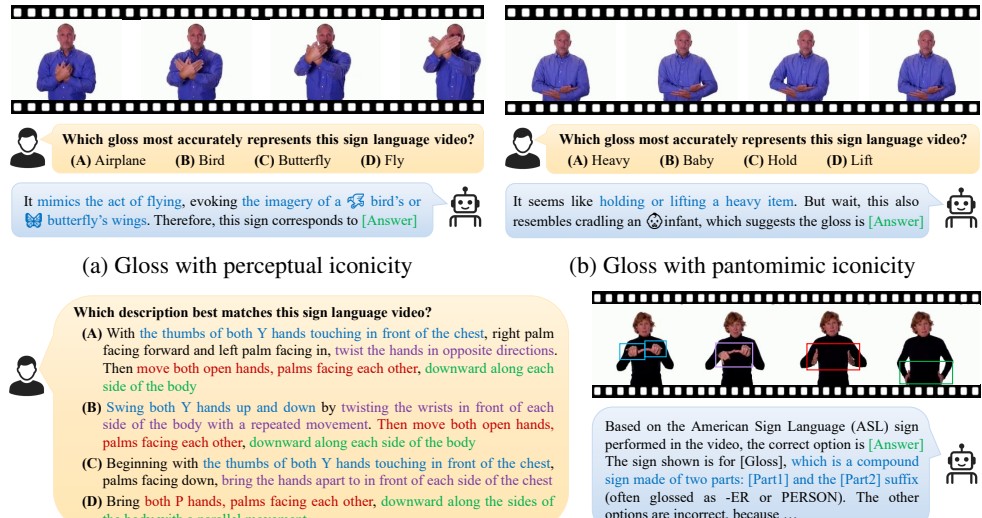

(a) Gloss with perceptual iconicity

(b) Gloss with pantomimic iconicity

(c) Examples of subunit-level perceptual questions based on lexical descriptions

Figure 1: Illustration of the LexSign-Bench questions and the corresponding answers[3], highlighting sign language iconicity types and the fine-grained structure within lexical descriptions.

the critical importance of capturing subunit-level structures, motivating our exploration of lexical descriptions to model these relationships. In this work, we broaden the concept of "lexical description" to encompass both lexical definitions provided in sign language dictionaries and automatically generated descriptive captions.

To better reveal the potential of lexical description, we first construct LexSign, a large-scale dataset comprising lexical descriptions collected from two ways: manually curated from sign language dictionaries and automatically generated through large foundation models. LexSign augments existing ISLR datasets using lexical descriptions for about 4,000 lexicons of different languages, broadening both the scale and scope of existing lexical definition datasets (Bilge et al., 2019; 2022). To guarantee the quality of generated descriptions, we build LexSign-Bench, a benchmark to comprehensively evaluate the sign language understanding capability of Multimodal Large Language Models (MLLMs), covering 300 glosses across different iconicity types. Based on the evaluation results, we further propose a perceive-then-summarize pipeline that leverages both the perception capability of MLLMs and the summarizing capability of LLMs to generate high-quality lexical descriptions, which improves the quality of generated descriptions in LexSign.

Different from general human action, sign language conveys meaning through explicit sequential and simultaneous composition of sign language subunits (Sandler & Lillo-Martin, 2006). To fully leverage the collected lexical description for advancing SLU, we propose Hierarchical Action-Language Interaction (HALI) that semantically aligns hierarchical visual representation with corresponding subunits captured in lexical description, thereby facilitating distinguishable and generalizable visual representation. Specifically, we first propose a multi-granularity contrastive loss to align visual and textual features of similar granularity, and incorporate a consistency constraint between visual representations at different levels to leverage the inherent hierarchical structure of sign language. Experimental results on public zero-shot and isolated datasets demonstrate the effectiveness of both the collected lexical descriptions and the proposed HALI framework.

In conclusion, this paper explores the potential of lexical descriptions for advancing SLU. The main contributions are summarized as follows:

- We construct LexSign, a high-quality and scalable dataset comprising lexical descriptions for approximately 4,000 sign language glosses.
- We develop LexSign-Bench, a benchmark for comprehensively evaluating the sign language understanding capability of MLLMs.

---

[3]The correct answers for questions (a), (b), and (c) are C, B, and A, respectively. The glosses corresponding to the choices for question (c) are "*Engineer*", "*Player*", "*Size*", and "*Person*".

- We propose HALI, a multi-granularity hierarchical alignment framework that fully utilizes lexical descriptions to obtain more distinguishable and generalizable visual representations.

## 2 RELATED WORK

### 2.1 ISOLATED SIGN LANGUAGE RECOGNITION

Isolated Sign Language Recognition (ISLR), which aims to recognize individual signs, serves as a fundamental task in sign language understanding. Recent works can be broadly divided into vision-based and language-assisted approaches, distinguished by whether language data is incorporated.

**Vision-based ISLR.** The central challenge of ISLR lies in effectively capturing distinguishable representations, and recent vision-based ISLR methods have advanced the field by leveraging cross-domain knowledge (Li et al., 2020b), employing self-supervised pre-training strategies (Hu et al., 2023; Zhao et al., 2023), and exploiting the intrinsic visual characteristics of sign language (Lin et al., 2024; Li et al., 2025b). For instance, Li et al. (2020b) promotes domain-invariant features and suppresses domain-specific features within continuous signs and isolated signs, thereby transferring cross-domain knowledge to improve ISLR. Inspired by the success of self-supervised learning in Natural Language Processing (NLP), BEST (Zhao et al., 2023) introduces a BERT-like pre-training framework tailored for sign language that operates on pose triplet units, demonstrating its effectiveness across various ISLR datasets. Different from these, VSNet (Li et al., 2025b) utilizes the linguistic characteristics of sign language from skeleton data through a joint fusion strategy and a self-attention model for visual symbol modeling, achieving significant ISLR performance without complex pre-training. These methods primarily focus on the design of the visual side without considering the linguistic information, limiting their generalizability and robustness.

**Language-assisted ISLR.** Linguistic data contains rich semantic information that can facilitate robust and generalizable visual representation learning, giving rise to recent advancements that incorporate linguistic information to improve ISLR (Wong et al., 2023; Zuo et al., 2023; Bilge et al., 2019; 2022). To improve the recognition of visually indistinguishable signs (VISigns), NLA-SLR (Zuo et al., 2023) proposes a language-aware label smoothing strategy and an inter-modality mixup technique based on the semantic embedding of glosses. However, they do not leverage finer-grained information, which can provide richer semantic context and capture subtle distinctions between VISigns. Several works utilize annotated phonological features to augment ISLR datasets (Tavella et al., 2022) and improve ISLR performance (Kezar et al., 2023a;b; 2025). Kezar et al. (2023a) and Kezar et al. (2023b) employ explicit, disentangled phonological features as supervisory signals to improve the visual representation, thereby boosting ISLR performance. Kezar et al. (2025) builds a knowledge graph ASLKG based on expert knowledge and trains neuro-symbolic models, yielding strong performance in ISLR. As revealed in Bilge et al. (2019; 2022), the textual definition of sign language lexicon can improve the generalization ability of sign language models, enabling zero-shot sign language recognition (ZSSLR) by grounding visual representation in a manually curated set of textual definitions. Different from these works, we use the collected lexical descriptions to evaluate MLLMs' sign language understanding and explore the potential of both manually curated and automatically generated descriptions in sign language recognition tasks.

### 2.2 MLLM FOR VISUAL UNDERSTANDING

The field of MLLMs has witnessed significant progress recently, giving rise to numerous state-of-the-art models that exhibit strong capabilities across various vision-language tasks (Wang et al., 2023; Yin et al., 2024; Zhang et al., 2024). Some models can accommodate video as an inherent input modality or in the form of multiple images, thereby enabling video understanding (Lin et al., 2023; Chen et al., 2024; Li et al., 2025a). For instance, Chen et al. (2024) collects a large-scale video caption dataset utilizing the proposed differential sliding-window captioning pipeline, and trains an image MLLM using video data to continually unlock its video understanding capability. Meanwhile, pre-existing datasets tailored for specific video understanding tasks are insufficient for a holistic and in-depth evaluation of MLLM's capabilities. Numerous MLLM evaluation benchmarks have been proposed (Yue et al., 2024; Xia et al., 2025; Zhou et al., 2025a; Hong et al., 2025) to address this limitation. For instance, MotionBench (Hong et al., 2025) is proposed to evaluate MLLM's motion-level perception capability, while MLVU (Zhou et al., 2025a) is proposed to evaluate MLLM's long

video understanding capability across different fields and tasks. Two recent works (Kim et al., 2025; Asasi et al., 2025) propose to automatically generate sign language descriptions for improving sign language translation. Different from these works, we improve the quality of generated descriptions by meticulously selecting the most capable MLLM through comprehensive evaluation, and further evaluate the quality of generated descriptions with the help of manually curated lexical descriptions.

### 2.3 LANGUAGE-ASSISTED ACTION RECOGNITION

Due to the greater accessibility of coarse-grained descriptions of human actions, numerous studies explore the effects of linguistic information in human action recognition (Wang et al., 2021; Ni et al., 2022; Ju et al., 2022; Pan et al., 2022; Rasheed et al., 2023; Liu et al., 2023) with the help of vision-language pre-training advances (Radford et al., 2021; Jia et al., 2021; Yao et al., 2022). By utilizing pre-defined prompt templates, ActionCLIP (Wang et al., 2021) extracts semantic representations of action labels, which are subsequently used to supervise visual representation learning and facilitate zero-shot action recognition. To capture subtle and discriminative motions inherent in complex human activities, which are essential for distinguishing visually similar actions, several studies employ LLMs to generate fine-grained descriptions of actions (Xiang et al., 2023; Jia et al., 2024; Bosetti et al., 2024; Liu et al., 2024; Zhu et al., 2024). For example, GAP (Xiang et al., 2023) leverages GPT-3 to generate textual descriptions of varying granularity through carefully designed prompts and proposes a multi-part contrastive learning framework to align visual and textual part features. To enable robust fine-grained alignment, PURLS (Zhu et al., 2024) introduces an adaptive partitioning approach that aggregates visual representations associated with local visual concepts extracted from GPT-3. In this work, we fully leverage the collected lexical description to advance SLU by conducting fine-grained semantic alignment between visual representation and lexical description, considering the sequentiality and simultaneity nature of sign language.

## 3 LEXSIGN

We first propose the lexical description collection method, and the resulting collected dataset LexSign in Sect. 3.1. Then, we propose the constructed LexSign-Bench for sign language understanding capability evaluation for MLLMs in Sect. 3.2.

### 3.1 CONSTRUCTION OF LEXSIGN

To better reveal the effectiveness of lexical descriptions, we first construct LexSign, a large-scale lexical description dataset of sign language. LexSign extends the WLASL (Li et al., 2020a) and DEVISIGN (Chai et al., 2014) datasets with lexical descriptions, resulting in LexSign-ASL and LexSign-CSL, respectively. As shown in Fig. 2, the lexical descriptions are collected via two data collection pipelines, the Manual Curation Pipeline (MCP) and the Automated Generation Pipeline (AGP). The resulting LexSign dataset includes descriptions obtained from both MCP and AGP for each gloss, enabling sampling from multiple sources to improve generalization. The demonstration of several examples obtained from MCP and AGP is provided in Supplementary Sect. A.3.

**Manual Curation Pipeline (MCP).** We manually extract lexical descriptions from sign language dictionaries, obtaining accurate lexical descriptions annotated by sign language experts. As illustrated in Fig. 2, an OCR tool is initially applied to convert the entire dictionary content into text, which is subsequently refined by a large language model to automatically correct potential OCR-induced errors. Next, lexical description candidates are retrieved from the processed text, with the gloss serving as the query. Finally, human annotators carefully verify the retrieved candidates by comparing them against the sign language video associated with the queried gloss, selecting the most accurate lexical description. Using the aforementioned method, we collect lexical descriptions paired with all 2,000 glosses in WLASL from Costello (2008) and 1,878 glosses in DEVISIGN from China Association of the Deaf (2003), respectively.

**Automated Generation Pipeline (AGP).** Although MCP produces accurate lexical descriptions curated by experts, it is inefficient or impractical for human annotators to retrieve signs not included in existing dictionaries. This highlights the need for an automated method that can effectively generate lexical descriptions at scale. To address this issue, we propose AGP, a fully automated lexical description generation pipeline in a perceive-then-summarize manner. As illustrated in Fig. 2, we first

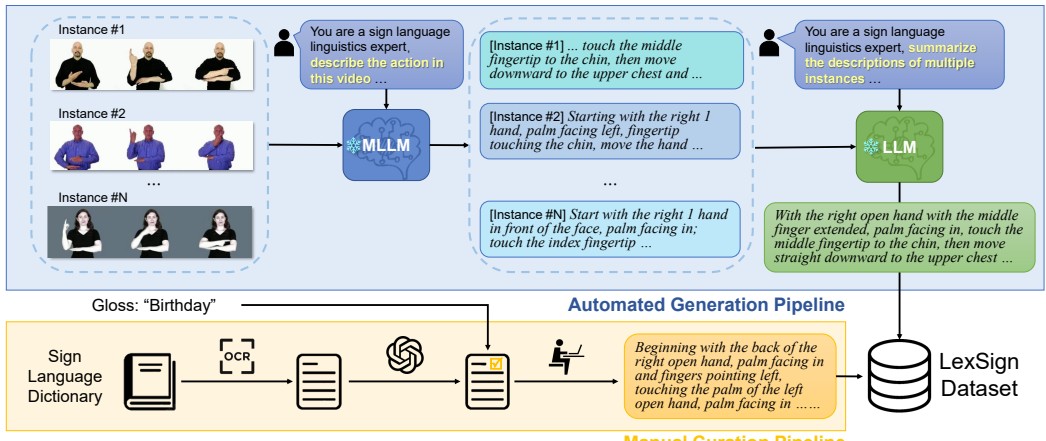

Figure 2: Illustration of MCP and AGP. For MCP, we first retrieve potential matching lexical descriptions from sign language dictionaries, and then meticulously select the best matching lexical description by human annotators. For AGP, an LLM summarizes the descriptions generated by MLLM for different videos associated with the same gloss to ensure consistency.

employ an MLLM to generate descriptions for individual videos corresponding to the same gloss. An LLM then aggregates these candidate descriptions to produce a final summarized description, enhancing cross-instance consistency while reducing intra-instance variability. Detailed prompts for large foundation models are provided in Supplementary Sect. A.14.

## 3.2 LEXSIGN-BENCH

While existing benchmarks have advanced MLLMs, a comprehensive evaluation of their sign language understanding remains limited. To address this, we construct LexSign-Bench. The evaluation tasks and scope, followed by the video collection and dataset construction process, are as follows.

**Tasks of LexSign-Bench.** We construct LexSign-Bench following a three-tiered scope: subunit-level perception, gloss-level recognition, and sentence-level translation. 1) **Subunit-level perception** (Fig. 1c) evaluates MLLMs' ability to identify subunits in a sign language video by selecting the correct lexical description from multiple-choice options; 2) **Gloss-level recognition** (Fig. 1a and Fig. 1b) assesses MLLMs' capability to recognize the sign language gloss in a sign language video using a similar multiple-choice setup; 3) **Sentence-level translation** measures the ability to translate sign language videos directly into natural language sentences. Preliminary experiments show that current MLLMs struggle with this task, so sentence-level translation is deferred to future work. Detailed prompts for MLLMs are provided in Supplementary Sect. A.14.

**Construction of LexSign-Bench.** We select glosses from the constructed LexSign-ASL considering their iconicity type (denoted as 'arbitrary', 'perceptual', 'pantomimic', and 'both', as shown in Fig. 4) based on the annotations in ASL-LEX (Sehyr et al., 2021). We select 75 glosses for each iconicity type in LexSign-ASL and collect all their video samples, resulting in a total of 300 glosses and 3,648 video samples. Two multiple-choice questions are curated per video, one for subunit-level perception evaluation and another for gloss-level recognition evaluation, resulting in a total of 7,296 multiple-choice questions with four choices per question. The correct answers are distributed approximately uniformly across all questions. To increase difficulty, we devise a hard-distractor mining strategy that selects the most confusable distractors identified from an ISLR model. Additional details are provided in Supplementary Sect. A.10.

## 4 METHOD

In this section, we present the proposed HALI, which consists of the multi-granularity alignment loss (Sect. 4.1) and the hierarchical consistency loss (Sect. 4.2).

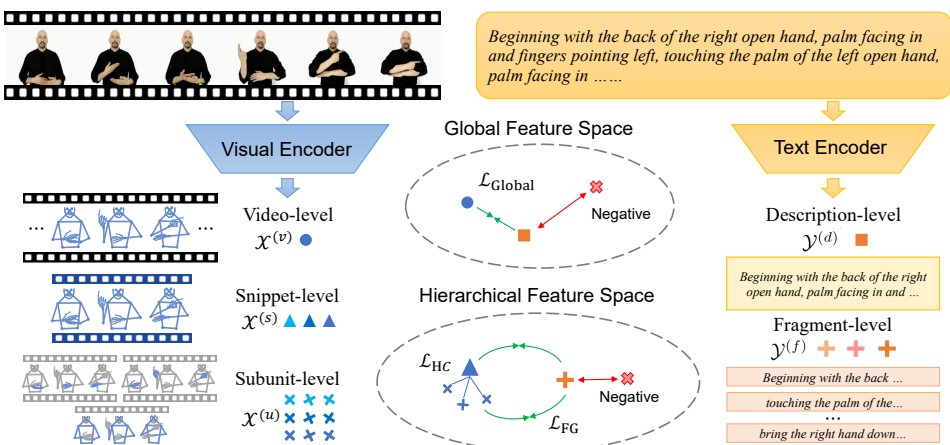

Figure 3: Illustration of the proposed HALI, which exploits the hierarchical structure of visual and lexical representations and provides supervision through both alignment and consistency constraints.

## 4.1 HIERARCHICAL ACTION-LANGUAGE ALIGNMENT

A sign language lexical description dataset $\mathcal{D} = \{(\mathcal{V}_i, \mathcal{S}_i)\}_{i=1}^N$ contains $N$ paired sign language video and description pairs. For notational convenience, we omit the subscript $i$ when it is clear from context. Given a sign language video $\mathcal{V} = \{\boldsymbol{v}^1, \cdots, \boldsymbol{v}^T\}$ with $T$ frames and its corresponding lexical description $\mathcal{S} = \{\boldsymbol{w}^1, \cdots, \boldsymbol{w}^L\}$ with $L$ tokens, we first extract their hierarchical visual and textual representation based on a vision encoder $E_v$ and a text encoder $E_t$ in Equ. 1.

$$\mathcal{X}^{(v)}, \mathcal{X}^{(s)}, \mathcal{X}^{(u)} = E_v(\mathcal{V}); \ \mathcal{Y}^{(d)}, \mathcal{Y}^{(f)} = E_t(\mathcal{S}). \tag{1}$$

$\mathcal{X}^{(v)} \in \mathbb{R}^{1 \times D}$, $\mathcal{X}^{(s)} \in \mathbb{R}^{T_s \times D}$, and $\mathcal{X}^{(u)} \in \mathbb{R}^{T_u \times D}$ are video-, snippet-, and subunit-level visual features, where $T_s$ and $T_u$ are the length of snippet- and subunit-level visual features, respectively, and $D$ is the feature dimension. $\mathcal{Y}^{(d)} \in \mathbb{R}^{1 \times D}$ and $\mathcal{Y}^{(f)} \in \mathbb{R}^{L_f \times D}$ are description- and fragment-level textual features, where $L_f$ is the number of fragments. Sign language conveys meaning through explicit sequential and simultaneous composition of sign language subunits. Therefore, an isolated sign can be temporally decomposed into a sequence of consecutive sub-actions, each of which can further be spatially decomposed into multiple subunits. We intend to align the snippet-level representation with a sub-action, the subunit-level representation with a subunit, and the fragment-level representation with either. Notably, each snippet-level feature corresponds to $M$ subunit-level features (e.g., left hand and right hand), yielding $T_u = M \times T_s$.

Lexical description can provide supervision through global vision-language alignment (Radford et al., 2021). For a mini-batch of $B$ sign language video and lexical description pairs $\{(\mathcal{V}_i, \mathcal{S}_i)\}_{i=1}^B$, we calculate the global contrastive loss following Equ. 2, where $\tau$ denotes the temperature.

$$\mathcal{L}_{\text{Global}} = -\frac{1}{2B} \sum_{i=1}^B \left( \log \frac{\exp(s_{v2d}(\mathcal{V}_i, \mathcal{S}_i)/\tau)}{\sum_{j=1}^B \exp(s_{v2d}(\mathcal{V}_i, \mathcal{S}_j)/\tau)} + \log \frac{\exp(s_{d2v}(\mathcal{S}_i, \mathcal{V}_i)/\tau)}{\sum_{j=1}^B \exp(s_{d2v}(\mathcal{S}_i, \mathcal{V}_j)/\tau)} \right). \tag{2}$$

The similarity between video- and description-level features is measured by cosine similarity $\rho(\cdot, \cdot)$:

$$s_{v2d}(\mathcal{V}, \mathcal{S}) = \rho(\mathcal{X}^{(v)}, \mathcal{Y}^{(d)}), \quad s_{d2v}(\mathcal{S}, \mathcal{V}) = \rho(\mathcal{Y}^{(d)}, \mathcal{X}^{(v)}). \tag{3}$$

Leveraging the constructed LexSign, we can further conduct fine-grained alignment between sign video and lexical description to obtain more discriminative visual features. Taking snippet-fragment alignment as an example, we calculate the affinity matrix $\mathcal{A}^{(s,f)} \in \mathbb{R}^{T_s \times L_f}$ between $\mathcal{X}^{(s)}$ and $\mathcal{Y}^{(f)}$ through cosine similarity, where $\mathcal{A}_{i,j}^{(s,f)}$ represents the similarity of the $i$-th video snippet and the $j$-th description fragment. Then, we calculate the fine-grained alignment (Yao et al., 2022) by:

$$s_{v2d}^{(s,f)}(\mathcal{V}, \mathcal{S}) = \frac{1}{T_s} \sum_{t=1}^{T_s} \max_l \left( \mathcal{A}_{t,l}^{(s,f)} \right); \ s_{d2v}^{(f,s)}(\mathcal{S}, \mathcal{V}) = \frac{1}{L_f} \sum_{l=1}^{L_f} \max_t \left( \mathcal{A}_{t,l}^{(s,f)} \right). \tag{4}$$

Notably, $s_{v2d}^{(s,f)}(\cdot,\cdot)$ represents the average similarity of each video snippet to its most relevant description fragment, with $s_{d2v}^{(f,s)}(\cdot,\cdot)$ defined analogously. The snippet-fragment fine-grained contrastive loss $\mathcal{L}_{\text{FG}}^{(s,f)}$ is computed by applying Equ. 4 within Equ. 2. Similarly, the subunit–fragment fine-grained loss $\mathcal{L}_{\text{FG}}^{(u,f)}$ can be obtained, yielding the multi-grained contrastive loss, where the loss weights of the fine-grained loss are $w_{\text{FG},s}$ and $w_{\text{FG},u}$:

$$\mathcal{L}_{\text{MG}} = \mathcal{L}_{\text{Global}} + w_{\text{FG},s}\mathcal{L}_{\text{FG}}^{(s,f)} + w_{\text{FG},u}\mathcal{L}_{\text{FG}}^{(u,f)}. \tag{5}$$

### 4.2 Action-Language Interaction Constraint

We further propose an alignment constraint loss that exploits the triplet relationship inherent in hierarchical structures (see Supplementary Sect. A.2) to directly encourage the consistency between the two fine-grained alignment results. Specifically, we first interpolate the subunit-fragment alignment matrix to match the shape of the snippet-fragment alignment matrix, respecting the inherent hierarchy between subunits and snippets, and then compute the hierarchical consistency loss as:

$$\mathcal{L}_{\text{HC}} = \frac{1}{T_s} \sum_{t=1}^{T_s} \left( D_{\text{KL}}(P_t^{(s,f)} \| P_t^{(u,f)}) + D_{\text{KL}}(P_t^{(u,f)} \| P_t^{(s,f)}) \right), \tag{6}$$

where $P^{(s,f)}$ and $P^{(u,f)}$ are similarity distributions derived from the corresponding alignment matrices, and $D_{KL}(\cdot\|\cdot)$ calculates the KL divergence. The final HALI loss is calculated following Equ. 7, where the loss weight of the hierarchical consistency loss is $w_{\text{HC}}$.

$$\mathcal{L}_{\text{HALI}} = \mathcal{L}_{\text{MG}} + w_{\text{HC}}\mathcal{L}_{\text{HC}}. \tag{7}$$

## 5 Experiment

### 5.1 Experimental Setup

**Datasets for ZSSLR Task.** We evaluate the proposed method on ASL-Text (Bilge et al., 2019), LexSign-ASL, and LexSign-CSL for the ZSSLR task under both zero-shot learning (ZSL) and generalized zero-shot learning (GZSL) settings in line with the protocol proposed by Xian et al. (2018a). ASL-Text comprises 250 American Sign Language glosses with a total of 1598 video samples. LexSign-ASL and LexSign-CSL contain 21,083 and 22,536 video samples with 2,000 and 1,878 glosses, respectively. For LexSign-ASL (and LexSign-CSL), we propose three different settings LexSign-ASL1000/300/100 (and LexSign-CSL1000/300/100) with a decreasing number of seen classes 1000/300/100 (and 983/299/100) and identical validation/test partitions with 400/600 (and 344/551) classes, enabling evaluation of the method's scalability.

**Datasets for ISLR Task.** We utilize WLASL (Li et al., 2020a) for the ISLR task, as the lexical annotations in LexSign-ASL are aligned with WLASL. WLASL is a large-scale, signer-independent resource for the ISLR task, comprising 21,083 video samples spanning 2,000 sign classes collected from educational platforms and YouTube tutorials. This dataset includes four progressively challenging subsets (WLASL-100/300/1000/2000), designed to evaluate model scalability.

**Implementation Details.** We perform pyramidal aggregation on the 1/2- and 1/4-scale frame features, producing a snippet-level representation of length 6 (=2+4) for each isolated sign. We also aggregate the left-hand, right-hand, and body features captured at these scales to construct the subunit-level representation, resulting in a subunit-level representation of length 18 (=3*6) for each isolated sign. We use most of the naturally occurring punctuation marks in the lexical descriptions, such as commas and periods, to segment each lexical description into fragments. Models trained solely with $\mathcal{L}_{\text{Global}}$ are the baseline for the ZSSLR task, while those trained solely with cross-entropy loss are the baseline for the ISLR task. The visual encoder is implemented as the CoSign-1s (Jiao et al., 2023). The text encoder is implemented as the BERT model (Devlin et al., 2019). All models are trained using the AdamW optimizer with a cosine annealing schedule. The learning rate and the number of training epochs for each task are provided in Supplementary Sect. A.1. We use a consistent experimental setup across datasets for the same task.

Table 1: Performance comparison (%) on ASL-Text. The best performance is highlighted in **bold**. Results marked with † are reproduced by (Bilge et al., 2022), while ⋆ indicates our reimplementation. As for GZSL, only the last six rows are comparable due to unavailable splits (Bilge et al., 2022). GZSL-S, GZSL-U, and GZSL-H represent the accuracies on seen classes, unseen classes, and their harmonic mean, respectively, in the GZSL setting.

| | ZSL | | GZSL-S | | GZSL-U | | GZSL-H | |
|---|---|---|---|---|---|---|---|---|
| | Top-1 | Top-5 | Top-1 | Top-5 | Top-1 | Top-5 | Top-1 | Top-5 |
| SAE† (Kodirov et al., 2017) | 8.0 | 16.0 | - | - | - | - | - | - |
| ESZSL† (Romera-Paredes & Torr, 2015) | 17.1 | 43.0 | - | - | - | - | - | - |
| f-CLSWGAN† (Xian et al., 2018b) | - | - | 33.3 | 64.6 | 6.7 | 20.7 | 11.1 | 31.3 |
| TF-VAEGAN† (Narayan et al., 2020) | - | - | 35.2 | 66.8 | 7.1 | 23.5 | 11.8 | 34.7 |
| LLE (Bilge et al., 2019) | 20.9 | 51.4 | - | - | - | - | - | - |
| LLE$_{Attr}$ (Bilge et al., 2022) | 23.7 | 59.2 | - | - | - | - | - | - |
| LLE$_{Attr+Text}$ (Bilge et al., 2022) | 31.3 | 66.0 | 37.0 | 72.4 | 5.5 | 20.3 | 9.5 | 31.7 |
| LLE (I3D)⋆ (Bilge et al., 2019) | 21.8 | 52.9 | 20.0 | 47.0 | 2.3 | 15.3 | 4.2 | 23.1 |
| LLE (CoSign)⋆ (Bilge et al., 2019) | 24.7 | 54.6 | **81.9** | 95.5 | 2.1 | 18.4 | 4.1 | 30.7 |
| DVTA (CoSign)⋆ (Kuang et al., 2025) | 17.0 | 47.2 | 19.6 | 61.0 | 2.6 | 12.4 | 4.6 | 20.4 |
| PGFA (CoSign)⋆ (Zhou et al., 2025b) | 26.8 | 51.9 | 62.2 | 86.9 | 2.1 | 11.2 | 4.1 | 19.9 |
| Baseline | 29.9 | 68.3 | 76.3 | 95.1 | 5.4 | 31.0 | 10.1 | 46.7 |
| Ours | **40.1** | **74.3** | 81.1 | **95.8** | **6.4** | **33.0** | **11.9** | **49.0** |

Table 2: Performance comparison (%) on LexSign-ASL. S, U, and H denote the accuracies on seen classes, unseen classes, and their harmonic mean, respectively.

| | LexSign-ASL100 | | | | LexSign-ASL300 | | | | LexSign-ASL1000 | | | |
|---|---|---|---|---|---|---|---|---|---|---|---|---|
| | ZSL | GZSL | | | ZSL | GZSL | | | ZSL | GZSL | | |
| | | S | U | H | | S | U | H | | S | U | H |
| LLE (S3D (Xie et al., 2018)) | 2.3 | 10.7 | 2.9 | 4.6 | 5.2 | 12.9 | 3.6 | 5.7 | 14.8 | 14.8 | 5.2 | 7.7 |
| LLE (I3D (Carreira & Zisserman, 2017)) | 2.0 | 10.2 | 3.0 | 4.6 | 5.8 | 14.5 | 3.6 | 5.8 | 15.6 | 14.7 | 5.8 | 8.3 |
| LLE (I3D (Varol et al., 2021)) | 3.2 | 36.6 | 4.5 | 8.0 | 7.5 | 40.0 | 5.3 | 9.4 | 20.3 | 37.0 | 7.7 | 12.8 |
| LLE (CoSign (Jiao et al., 2023)) | 3.1 | **49.6** | 5.4 | 9.8 | 8.2 | **50.2** | 6.1 | 10.9 | 21.4 | **46.7** | 8.5 | 14.4 |
| Baseline | 4.1 | 33.1 | 6.1 | 10.4 | 9.2 | 33.4 | 7.0 | 11.6 | 21.0 | 28.0 | 7.6 | 11.9 |
| Ours | **4.7** | 43.3 | **7.2** | **12.3** | **13.6** | 45.0 | **9.0** | **15.0** | **27.6** | 41.3 | **11.2** | **17.6** |

**Evaluation Metrics.** Unless stated otherwise, experiments use lexical descriptions extracted via MCP. For ISLR, we report per-class and per-instance top-1 accuracy. For ZSSLR, we report per-class top-1/top-5 accuracy in the ZSL setting, and per-class top-1/top-5 accuracy for seen and unseen classes, along with their harmonic mean, in the GZSL setting.

## 5.2 RESULTS

**Evaluation Results on LexSign-Bench** We evaluate three closed-source MLLMs (GPT-5 (OpenAI, 2025), Gemini 2.5 Pro (Comanici et al., 2025), Qwen-VL-Max (Bai et al., 2023)), and three open-source MLLMs (InternVL3.5 (Wang et al., 2025), Qwen2.5-VL (Bai et al., 2025), LLaVA-OneVision (Li et al., 2025a)), which are representative state-of-the-art models and therefore provide reasonable and reliable baselines for LexSign-Bench. The results across evaluation tasks and iconicity types are shown in Fig. 4, and the detailed numbers are provided in Supplementary Sect. A.10. Among all the evaluated MLLMs, GPT-5 achieves the best performance (average 65.0% accuracy), surpassing the second-ranked Gemini by 6.2%. Closed-source MLLMs consistently outperform 7/8B open-source models, suggesting that the latter lack sufficient sign language expertise, likely due to limitations in training data and model size. Except for InternVL3.5, all evaluated MLLMs achieved higher accuracy on iconic signs than on arbitrary ones, corroborating our hypothesis that iconicity is positively correlated with comprehensibility. Notably, for the two leading MLLMs, GPT-5 and Gemini 2.5 Pro, their performance on gloss-level tasks surpasses that on subunit-level tasks, which stands in contrast to the conclusions drawn for other evaluated models. This observation suggests that GPT-5 and Gemini 2.5 Pro can correctly recognize glosses without precisely capturing fine-grained articulatory details. In summary, recent MLLMs demonstrate a degree of sign language expertise and show potential for use in sign language understanding tasks.

**User Study for LexSign-Bench.** To provide a baseline for MLLM performance on LexSign-Bench, we conduct a user study. Specifically, 10 signers and 10 non-signers completed a subset of LexSign-Bench through a questionnaire consisting of 32 questions, resulting in two separate benchmark re-

Table 3: Performance comparison (%) on LexSign-CSL. S, U, and H denote the accuracies on seen classes, unseen classes, and their harmonic mean, respectively.

| | LexSign-CSL100 | | | | LexSign-CSL300 | | | | LexSign-CSL1000 | | | |
| | ZSL | GZSL | | | ZSL | GZSL | | | ZSL | GZSL | | |
| | | S | U | H | | S | U | H | | S | U | H |
|---|---|---|---|---|---|---|---|---|---|---|---|---|
| LLE (S3D (Xie et al., 2018)) | 1.6 | 31.9 | 2.6 | 4.8 | 4.4 | 31.9 | 3.6 | 6.5 | 11.1 | 30.7 | 4.7 | 8.1 |
| LLE (I3D (Carreira & Zisserman, 2017)) | 1.7 | 28.6 | 2.9 | 5.3 | 4.6 | 27.3 | 3.6 | 6.3 | 11.9 | 26.8 | 5.7 | 9.3 |
| LLE (I3D (Varol et al., 2021)) | 2.4 | 61.6 | 4.6 | 8.5 | 6.4 | 63.0 | 5.8 | 10.7 | 16.1 | 58.9 | 7.3 | 12.9 |
| LLE (CoSign (Jiao et al., 2023)) | 2.8 | **79.5** | 5.7 | 10.6 | 7.3 | **76.9** | 7.2 | 13.2 | 18.2 | **70.2** | 10.5 | 18.3 |
| Baseline | 4.1 | 70.1 | 11.1 | 19.2 | 16.1 | 66.8 | 14.4 | 23.6 | 30.7 | 60.8 | 18.9 | 28.9 |
| Ours | **5.4** | 74.7 | **12.8** | **21.8** | **18.4** | 72.4 | **16.7** | **27.1** | **36.2** | 68.2 | **21.3** | **32.5** |

Table 4: Top-1 accuracy (%) on WLASL. P-I / P-C correspond to per-instance / per-class results.

| | WLASL100 | | WLASL300 | | WLASL1000 | | WLASL2000 | |
| | P-I | P-C | P-I | P-C | P-I | P-C | P-I | P-C |
|---|---|---|---|---|---|---|---|---|
| *RGB-based* | | | | | | | | |
| SignBERT+ (Hu et al., 2023) | 84.11 | 85.05 | 78.44 | 79.12 | - | - | 55.59 | 53.33 |
| NLA-SLR (Zuo et al., 2023) | 92.64 | 93.08 | 86.98 | 87.33 | 75.64 | 75.72 | 61.26 | 58.31 |
| Uni-Sign (Li et al., 2025c) | 92.25 | 92.67 | 88.47 | 88.92 | - | - | 63.52 | 61.32 |
| *Pose-based* | | | | | | | | |
| BEST (Zhao et al., 2023) | 77.91 | 77.83 | 67.66 | 68.31 | - | - | 46.25 | 43.52 |
| VSNet (Li et al., 2025b) | 85.66 | 86.25 | 80.09 | 80.85 | - | - | 55.98 | 53.54 |
| MSLU (Zhou et al., 2025c) | **88.76** | **89.25** | 82.04 | 82.71 | - | - | 56.29 | 53.29 |
| Baseline | 85.27 | 85.28 | 82.14 | 82.53 | 72.47 | 72.46 | 57.59 | 54.95 |
| Baseline + $\mathcal{L}_{Global}$ | 86.31 | 86.47 | 83.08 | 83.29 | 74.00 | **73.92** | 60.01 | 57.47 |
| Baseline + $\mathcal{L}_{HALI}$ | 86.43 | 86.89 | **83.73** | **84.19** | **74.07** | 73.86 | **60.68** | **58.32** |

sults, as listed in Supplementary Sect. A.10. It can be observed that, except for gloss-level questions where non-signers score lower than GPT-5, humans consistently score well above the MLLMs. In general, the human scores, particularly those from the signers, serve as the performance upper bound for MLLMs on LexSign-bench, showing that the models still have significant room to improve.

**ZSSLR Result.** As demonstrated in Table 1, in ASL-Text, introducing the HALI loss leads to a 10.2% performance improvement over the baseline, and outperforms LLE$_{Attr+Text}$ (Bilge et al., 2022) by 8.8% without requiring extra attribute annotations. Similar trends can be observed in LexSign, as demonstrated in Table 2 and Table 3. These results demonstrate HALI's effectiveness in utilizing hierarchical semantic structure within lexical descriptions. Besides, in LexSign, the proposed method consistently improves the recognition accuracy as the training classes increase, highlighting its scalability. It's worth noting that as the number of classes in the training set increases, the difficulty of the GZSL setting also rises and may lead to a decrease in seen class accuracy. Overall, HALI achieves state-of-the-art performance across all evaluated datasets on the ZSSLR task.

**ISLR Result.** We evaluate our method on the ISLR task to further assess the potential of lexical descriptions for advancing sign language understanding tasks. We leverage a strong baseline where the visual encoder is pre-trained on OpenASL and How2Sign datasets. As presented in Table 4, incorporating the HALI loss into the baseline improves top-1 per-instance and per-class accuracy by 3.09% and 3.37% on WLASL2000, respectively. Most of the performance gain comes from $\mathcal{L}_{Global}$, highlighting the quality of the collected lexical descriptions and their effectiveness under the supervised setting. The additional improvement from HALI on the challenging WLASL2000 highlights the benefit of modeling the subunit-level structure of sign language.

## 5.3 ABLATION STUDY

**Ablation on the Quality of Generated Lexical Descriptions with MLLMs.** We use MLLMs to generate descriptions for all glosses in LexSign-ASL, pairing each gloss with a single video. This process relies solely on the MLLMs' perception and captioning capabilities. To evaluate description quality, we train a ZSSLR model on the automatically generated descriptions and test it using manually collected lexical descriptions from sign language dictionaries. As shown in Table 5, experiments on LexSign-ASL1000 show that GPT-5-generated descriptions yield the best generalization, outperforming Qwen-VL-Max by 2.6% (top-1) and 8.2% (top-5), and substantially surpasses open-source models, consistent with the evaluation results on LexSign-Bench.

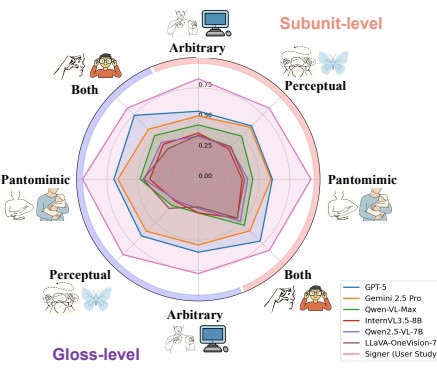

Figure 4: Evaluation results of MLLMs on the LexSign-Bench. Sign language diagrams are adapted from Sternberg (1995).

Table 5: Ablation (%) of MLLMs for lexical description acquisition on LexSign-ASL1000.

|  | Val | | Test | |
| --- | --- | --- | --- | --- |
|  | Top-1 | Top-5 | Top-1 | Top-5 |
| InternVL3.5-8B | 0.9 | 4.7 | 1.3 | 4.1 |
| Qwen2.5-VL-7B | 1.5 | 6.8 | 1.1 | 4.6 |
| Qwen-VL-Max | 2.9 | 10.2 | 2.1 | 8.5 |
| GPT-5 | 6.1 | 20.8 | 4.7 | 16.7 |

Table 6: Ablation (%) of summary strategy for lexical description acquisition on LexSign-ASL1000.

| #Instances | Qwen-VL-Max | | GPT-5 | |
| --- | --- | --- | --- | --- |
|  | Top-1 | Top-5 | Top-1 | Top-5 |
| 1 w/o Summarizing | 2.1 | 8.5 | 4.7 | 16.7 |
| 2 | 2.2 | 7.4 | 4.6 | 16.3 |
| 3 | 2.5 | 9.2 | 5.9 | 18.4 |
| 3 w/ Sampling | 2.3 | 9.3 | 6.7 | 22.1 |

Table 7: Ablation (%) of HALI on LexSign. T-1 / T-5 denote Top-1 / Top-5 accuracies, respectively.

|  |  |  | LexSign-ASL1000 | | | | LexSign-CSL1000 | | | |
| --- | --- | --- | --- | --- | --- | --- | --- | --- | --- | --- |
|  |  |  | Val | | Test | | Val | | Test | |
| (s,f) | (u,f) | $\mathcal{L}_{HC}$ | T-1 | T-5 | T-1 | T-5 | T-1 | T-5 | T-1 | T-5 |
|  |  |  | 26.5 | 54.2 | 21.0 | 44.2 | 39.9 | 68.7 | 30.7 | 59.7 |
| ✓ |  |  | 32.4 | 59.8 | 24.6 | 49.7 | 42.3 | 67.9 | 33.4 | 58.7 |
|  | ✓ |  | 29.7 | 57.1 | 22.8 | 48.5 | 42.8 | 69.6 | 33.7 | 60.4 |
| ✓ | ✓ |  | 34.2 | 61.1 | 25.7 | 51.3 | 43.8 | 70.6 | 35.2 | 60.3 |
| ✓ | ✓ | ✓ | **35.3** | **63.8** | **27.6** | **53.1** | **44.9** | **72.6** | **36.2** | **62.0** |

**Ablation on Summarizing Multiple Descriptions with LLMs.** As mentioned in Sect. 3.1, we use LLMs to aggregate MLLM-generated descriptions from multiple videos of the same gloss. To evaluate summarization, we conduct experiments on LexSign-ASL1000, replacing training descriptions with the summarized versions. We also evaluate the simple random sampling strategy from multi-source generated results, including before and after summarizing. As presented in Table 6, aggregating multiple generated descriptions yields a substantial improvement in quality, highlighting the LLM's capability to capture common patterns while accommodating individual variations. Additionally, for GPT-5, the proposed sampling strategy delivers a notable performance enhancement. We adopt GPT-5 as the default MLLM in AGP, considering its superior performance. However, the performance gap with descriptions from the dictionary still exists (6.7% *vs.* 27.6%), indicating considerable room for generating high-quality descriptions with an automatic pipeline.

**Ablation on Hierarchical Action-language Interaction.** As shown in Table 7, both snippet-fragment and subunit-fragment alignment yield significant performance gains, and combining multi-granularity alignments provides further improvements, highlighting their complementary roles. The proposed hierarchical consistency loss additionally boosts performance across datasets, underscoring the importance of high-quality alignment.

## 6 CONCLUSION

This paper focuses on collecting, generating, and leveraging lexical descriptions for advancing Sign Language Understanding (SLU) by capturing the subunit structure of sign language. Specifically, we first construct LexSign, a large-scale dataset that extends existing resources with high-quality lexical descriptions. Next, we introduce LexSign-Bench to comprehensively evaluate the sign language understanding capabilities of MLLMs, showing that recent models exhibit a degree of sign language expertise and potential for SLU tasks. Based on the collected data, we propose Hierarchical Action-Language Interaction (HALI), which performs multi-granularity hierarchical alignment between lexical descriptions and sign language videos. Experimental results verify the effectiveness of both the constructed lexical datasets and the proposed method. We hope that our dataset and approach will inspire future research in leveraging MLLMs for SLU and in designing more generalized models guided by linguistic information.

## 7 ETHICS STATEMENT

This research on sign language understanding is conducted with a deep commitment to ethical principles, prioritizing respect for the Deaf and Hard of Hearing communities. This work is intended to be a positive contribution to the field, fostering greater accessibility and understanding of sign language, such as open-vocabulary sign language translation. The LexSign dataset involves copyrighted sign language dictionaries, and we plan to provide the corresponding page numbers and locations in the dictionaries for all the collected lexical descriptions. All sign language videos are obtained from the existing datasets (Li et al., 2020a; Chai et al., 2014), and we leverage the estimated skeleton as input to ensure signer privacy by excluding personally identifiable information. We have taken care to ensure our methods and results are unbiased, with the goal of supporting and benefiting the Deaf community.

## 8 REPRODUCIBILITY STATEMENT

To ensure the full reproducibility of our research, we will make all key components of this work publicly available upon paper acceptance. As for the submitted version, we provide a detailed description of the experimental setup in Sect. 5.1 and Supplementary Sect. A.1. To guarantee the reproducibility of the data collection process, we thoroughly describe the data collection pipeline in Sect. 3.1 and Supplementary Sect. A.14. We will release LexiSign, encompassing LexSign-ASL, LexSign-CSL, and LexSign-Bench, with careful attention to copyright considerations.

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

Table 8: Learning rate and training epochs in the ZSSLR and ISLR tasks. Visual, Text, and Other denote the parameters of visual encoder, text encoder, and all remaining components, respectively.

| | Learning rate | | | Epoch |
|---|---|---|---|---|
| | Visual | Text | Other | |
| ISLR | $1 \times 10^{-4}$ | $1 \times 10^{-4}$ | $1 \times 10^{-3}$ | 80 |
| ZSSLR | $3 \times 10^{-5}$ | $1 \times 10^{-5}$ | $6 \times 10^{-5}$ | 40 |

Figure 5: Illustration of the collected descriptions generated by different pipelines.

## A APPENDIX

### A.1 OTHER IMPLEMENTATION DETAILS

Through experiments on ZSSLR with the LLE approach, we evaluated various visual encoders and observed that CoSign-1s (Jiao et al., 2023) generally yields better performance. Thus, in our baseline, we use CoSign-1s pre-trained on PHOENIX14T (Camgoz et al., 2018) as the visual encoder for the ZSSLR task, and pre-trained on OpenASL (Shi et al., 2022) and How2Sign (Duarte et al., 2021) for the ISLR task. Notably, PHOENIX14T is a DGS dataset, ensuring that no data leakage occurs when transferring CoSign-1s pre-trained on PHOENIX14T to an ASL or CSL task.

The learning rate and the number of training epochs for each task are provided in Table 8. The temperature $\tau$ used in Equ. 2 is set to 0.03. We trained skeleton-based models on ASL-Text (Bilge et al., 2019) and WLASL (Li et al., 2020a) three times and averaged their results.

### A.2 THE ASSUMPTION OF A HIERARCHICAL RELATIONSHIP AMONG MULTI-GRAINED VISUAL REPRESENTATIONS

While multi-grained action-language interaction proposed in Equ. 5 facilitates fine-grained alignment, it overlooks the potential hierarchical relationship between multi-grained visual representations. Specifically, we can assume that the same fragment of lexical description should attend to visual representations within the same snippet, which can be formulated as:

$$\rho(\mathcal{Y}_i^{(f)}, \mathcal{X}_j^{(s)}) \approx \frac{1}{M} \sum_{m=1}^{M} \rho(\mathcal{Y}_i^{(f)}, \mathcal{X}_m^{(u)}), \tag{8}$$

### A.3 QUALITATIVE RESULTS

**Examples for Collected Lexical Descriptions.** We provide three examples of glosses *Birthday*, *Watermelon*, and *World* to demonstrate the lexical descriptions collected through the Manual Curation Pipeline (MCP) and Automated Generation Pipeline (AGP), as shown in Fig. 5.

**Example for LexSign-Bench.** LexSign-Bench covers both subunit-level and gloss-level questions. For clarity, we provide a precise gloss-level question example, along with the output produced by GPT-5, as shown in Fig. 6. The original WLASL video is shown on the left, while the right videos are under temporal or spatial perturbations from the original video, refer to Supplementary Sect. A.8.

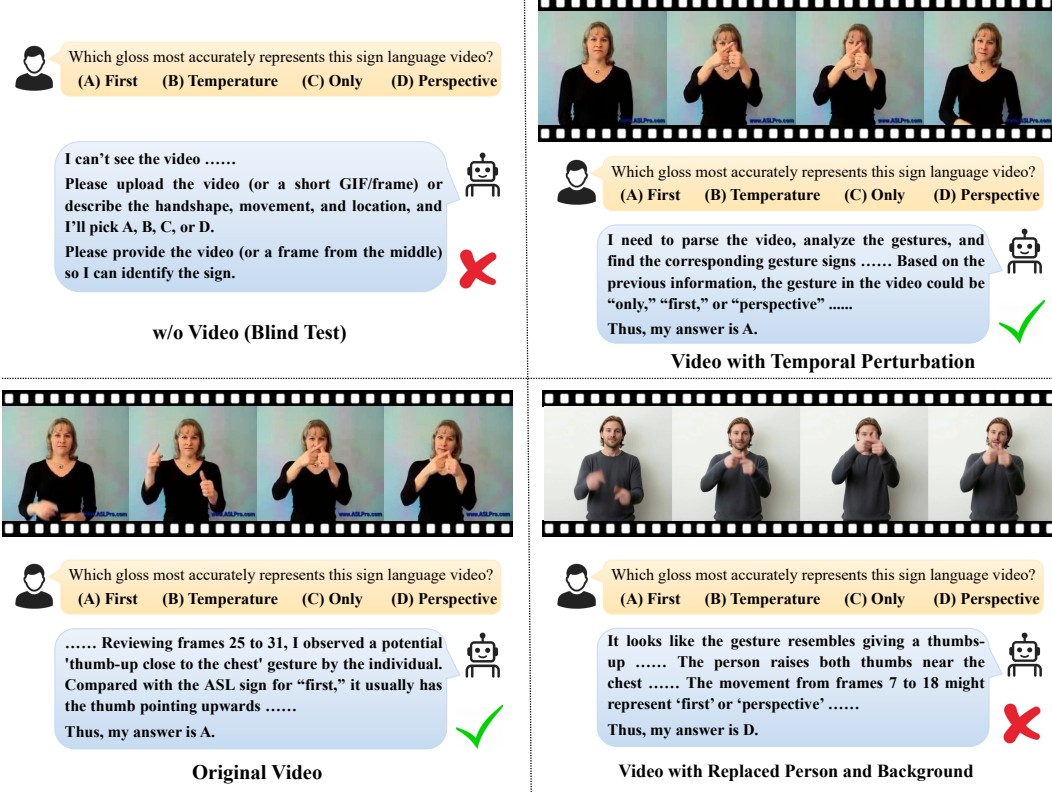

Figure 6: An example of a gloss-level question in LexSign-Bench and the corresponding output generated by GPT-5 with reasoning. GPT-5 answers correctly on the original WLASL video (left), but fails on the generated version with a replaced signer and background (right).

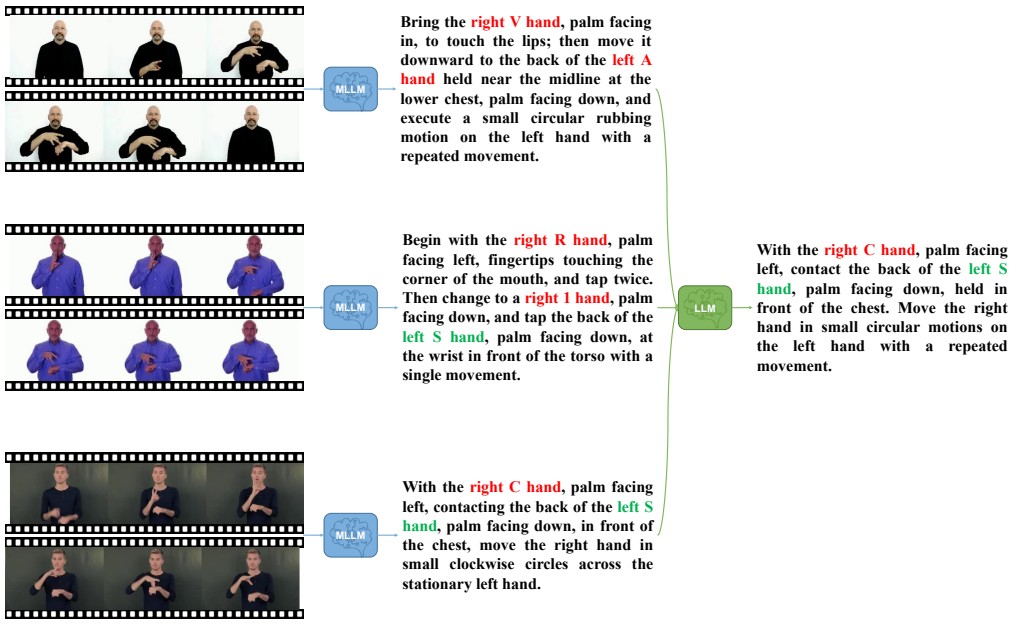

Figure 7: An example of a lexical description of gloss *Watermelon* collected from the Automated Generation Pipeline (AGP) using GPT-5. Red marks denote incorrect hand shapes, and green marks denote correct ones.

Table 9: Results (%) of ZSSLR on the LexSign-ASL1000 dataset using various fragment settings.

| | Val | | Test | |
|---|---|---|---|---|
| | Top-1 | Top-5 | Top-1 | Top-5 |
| Fragments generated by Gemini 2.5 Flash | 34.4 | 63.2 | 26.4 | 52.4 |
| Fragments generated by Qwen-Plus | 35.3 | 62.6 | 26.5 | 52.0 |
| Punctuation-based fragments | **35.3** | **63.8** | 27.6 | 53.1 |

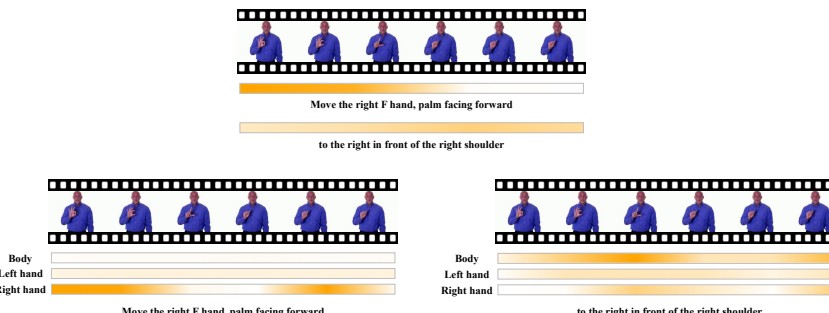

Figure 8: Qualitative results of fine-grained action–language alignments. Each horizontal bar illustrates the cosine similarity between the visual features and the textual features, with deeper colors indicating higher semantic similarity.

**Example for Description Collected from Automated Generation Pipeline.** We provide an example of a lexical description of gloss *Watermelon* collected from the Automated Generation Pipeline (AGP) using GPT-5, as demonstrated in Fig. 7. The generated descriptions are generally accurate, with most errors arising from the depiction of hand shapes. Overall, GPT-5 is capable of recognizing general hand movements related to the semantics of the sign.

**Qualitative Result for HALI.** To qualitatively evaluate the hierarchical action-language alignment quality, Fig. 8 depicts the predicted cosine similarity between visual features and textual features for an example in the validation set of LexSign-ASL. The upper part of Fig. 8 illustrates the alignment between fragment-level textual features and snippet-level visual features, while the lower left and right part of Fig. 8 visualizes the alignment between two fragments and subunit-level visual features, respectively. It can be observed that the first fragment attends primarily to the earlier snippet-level visual features, particularly the subunit-level feature of the right hand, whereas the second fragment focuses on the later snippet-level visual features. These results are consistent with their semantic content and the intrinsic hierarchical structure of the sign, which demonstrates the effectiveness of the proposed method.

### A.4 THE SEMANTIC SIGNIFICANCE OF FRAGMENT

By looking at examples from the dataset, we find clear evidence that fragments produced mainly through simple punctuation-based segmentation from lexical descriptions still carry meaningful semantic significance. In addition, we conduct a dimensionality-reduction visualization to inspect the clustering patterns of all fragments, as shown in Fig. 9. Specifically, we extract sentence embeddings for all fragments using BERT and apply t-SNE for dimensionality reduction. The resulting clusters show that similar fragments group together and describe similar sub-actions, further supporting the claim that these fragments contain semantic significance.

Moreover, we conduct an experiment in which we used an LLM to extract potential sub-action descriptions from the original lexical descriptions to replace the punctuation-based fragments. We then performed ZSSLR experiments on LexSign-ASL1000 using different fragments, and the results are shown in Table 9. It is observed that the fragments generated by the LLM lead to inferior performance compared to our simple punctuation-based fragments, further demonstrating the effectiveness of our rule-based splitting approach.

Table 10: Per-instance accuracy (%) of ISLR with CoSign-1s pretrained on different datasets.

| | PHOENIX14T | | How2Sign | | OpenASL+How2Sign | |
|---|---|---|---|---|---|---|
| | T-1 | T-5 | T-1 | T-5 | T-1 | T-5 |
| Baseline | 51.58 | 83.43 | 52.21 | 84.33 | 57.59 | 88.0 |
| Baseline + $\mathcal{L}_{\text{Global}}$ | 55.64 | 86.84 | 55.54 | 86.91 | 60.01 | 90.34 |
| Baseline + $\mathcal{L}_{\text{HALI}}$ | **55.85** | **88.05** | **56.03** | **88.12** | **60.68** | **90.93** |

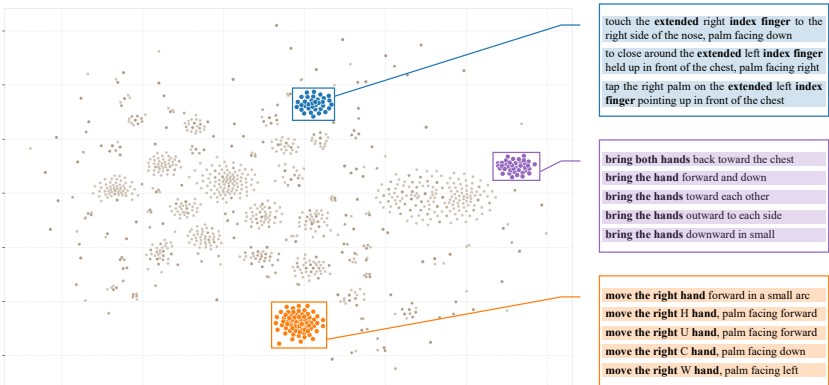

Figure 9: A t-SNE visualization of the embeddings extracted by BERT from all fragments obtained by splitting all lexical descriptions in LexSign-ASL. Representative examples from each color-matched cluster are shown in the boxes on the right.

## A.5 ABLATION ON THE PRETRAINED DATASET FOR THE VISUAL ENCODER

In the experiments presented in Table 4, the implemented CoSign-1s is pretrained on the OpenASL+How2Sign dataset. We further report the ISLR results for CoSign-1s pretrained on PHOENIX14T and How2Sign, as shown in Table 10. It can be observed that replacing the pretrain dataset from PHOENIX14T to How2Sign yields a modest performance improvement, which can be attributed to the fact that How2Sign and WLASL are based on the same language. Pretraining on How2Sign as well as on the large-scale dataset OpenASL leads to a significant performance boost. Notably, HALI consistently yields performance gains, demonstrating its generalizability.

## A.6 THE GENERALIZABILITY OF HALI

The global contrastive loss $\mathcal{L}_{\text{Global}}$ in HALI is flexible and can be substituted with other ZSL losses. We additionally experimented with two ZSL losses (Zhou et al., 2025b; Kuang et al., 2025) on LexSign-ASL1000 and LexSign-CSL1000, and observed that incorporating the proposed fine-grained loss $\mathcal{L}_{\text{FG}}$ and hierarchical consistency loss $\mathcal{L}_{\text{HC}}$ consistently and significantly boosts performance, highlighting the generalizability of HALI. Detailed results are presented in Table 11.

## A.7 FINETUNE THE MLLM WITH THE LEXSIGN DATASET

We finetune the MLLM using the LexSign dataset and observed that the resulting finetuned model produced substantially higher-quality descriptions. Specifically, we finetune Qwen2.5-VL using the video-description pairs of 700 glosses, which are present in LexSign-ASL1000 but absent from LexSign-ASL300, thereby ensuring that no data leakage occurs for the downstream task. We then apply AGP with the finetuned MLLM to generate descriptions for the training set of LexSign-ASL300. Based on these automatically generated descriptions, we trained a ZSSLR model and evaluated it using manually collected lexical descriptions from sign language dictionaries. All downstream evaluations are carried out on LexSign-ASL300, with results summarized in Table 12.

It is observed that the ZSSLR model trained with descriptions generated by the finetuned Qwen2.5-VL achieves a substantial performance improvement over the model trained with descriptions generated by the original model. This result confirms that LexSign is a valuable resource for finetuning MLLMs to improve their sign language understanding capability.

Table 11: Performance comparison (%) with different baselines on LexSign.

| | | LexSign-ASL1000 | | | | LexSign-CSL1000 | | | |
| | | Val | | Test | | Val | | Test | |
| | $\mathcal{L}_{FG} + \mathcal{L}_{HC}$ | T-1 | T-5 | T-1 | T-5 | T-1 | T-5 | T-1 | T-5 |
|---|---|---|---|---|---|---|---|---|---|
| DVTA (Kuang et al., 2025) | | 21.3 | 46.8 | 16.1 | 37.6 | 16.8 | 43.2 | 10.8 | 33.5 |
| DVTA (Kuang et al., 2025) | ✓ | 32.5 | **60.8** | 25.2 | 51.0 | 37.9 | **65.8** | 29.1 | 56.8 |
| PGFA (Zhou et al., 2025b) | | 32.1 | 53.3 | 25.1 | 43.2 | 39.4 | 59.1 | 29.8 | 48.3 |
| PGFA (Zhou et al., 2025b) | ✓ | **34.8** | 56.2 | 28.1 | 46.8 | **42.0** | 61.7 | 31.9 | 50.6 |

Table 12: Ablation (%) of MLLMs for lexical description acquisition on LexSign-ASL300.

| | Val | | Test | |
| | Top-1 | Top-5 | Top-1 | Top-5 |
|---|---|---|---|---|
| *Closed-source MLLMs* | | | | |
| GPT-5 | 9.7 | 29.0 | 6.7 | 22.1 |
| Qwen-VL-Max | 2.9 | 13.1 | 2.3 | 9.3 |
| *Open-source MLLMs* | | | | |
| Qwen2.5-VL-7B | 1.6 | 6.0 | 0.9 | 4.3 |
| Qwen2.5-VL-7B (Finetuned) | 2.9 | 10.2 | 1.6 | 6.5 |

Table 13: Results (%) for evaluating potential data leakage of MLLMs on LexSign-Bench. AVG (S) and AVG (G) correspond to the mean performance on subunit-level and gloss-level questions. Temp. Pert. and Spat. Pert. denote the results on sign videos under temporal perturbation and spatial perturbation, respectively.

| MLLM | Blind Test | | | Temp. Pert. | | | Spat. Pert. | | |
| | AVG | AVG (S) | AVG (G) | AVG | AVG (S) | AVG (G) | AVG | AVG (S) | AVG (G) |
|---|---|---|---|---|---|---|---|---|---|
| *Closed-source MLLMs* | | | | | | | | | |
| GPT-5 | 25.2 | 26.7 | 23.7 | 57.0 | 54.3 | 59.7 | 58.0 | 55.7 | 60.3 |
| *Open-source MLLMs* | | | | | | | | | |
| InternVL3.5-8B | 27.0 | 27.4 | 26.6 | 36.2 | 39.3 | 33.0 | 34.0 | 38.2 | 29.8 |
| Qwen2.5-VL-7B | 27.6 | 27.6 | 27.7 | 35.4 | 39.3 | 31.5 | 33.8 | 37.7 | 30.0 |
| LLaVA-OneVision-7B | 24.9 | 24.5 | 25.2 | 37.2 | 39.9 | 34.6 | 34.7 | 37.8 | 31.6 |

## A.8 POTENTIAL DATA LEAKAGE RISKS IN LEXSIGN-BENCH

To further validate LexSign-Bench and exclude potential data leakage effects, we introduce three additional tests: (1) a blind test, where only the text is provided without the corresponding video; (2) a temporal perturbation test, which manipulates the video sequence by randomly dropping or repeating a small number of frames with a probability of 0.2; and (3) a spatial perturbation test, where the signer and background are replaced by Wan2.2-Animate (Wan et al., 2025). Examples of the perturbed video instances are provided in Supplementary Sect. A.3. As shown in Table 13, blind-test results remain close to the 25% random-chance accuracy (each question has 4 choices), indicating that our benchmark is unbiased at the text-level. As for the video side, introducing temporal or spatial perturbations leads to a slight performance drop, which we attribute to the injected noise by temporal augmentation and diffusion models. However, the overall trend remains largely consistent, suggesting that current MLLMs do not benefit from prior exposure to the public datasets. GPT-5 maintains a clear advantage over all open-source MLLMs.

## A.9 PERFORMANCE OF CONVENTIONAL DEEP LEARNING MODEL ON LEXSIGN-BENCH

To provide a reference for MLLM's performance on LexSign-Bench, we evaluate conventional deep learning models trained on LexSign on this benchmark. Specifically, we adopt two setups. At the subunit level, a ZSSLR model is initially trained on a dataset comprising 1,500 glosses (which do not overlap with the 300 glosses in the LexSign-Bench), and is subsequently tasked with selecting the most similar lexical description from four candidates for each video in LexSign-Bench. At the gloss level, an ISLR model is first trained on WLASL2000, and then evaluated on the 300 glosses included in LexSign-Bench to identify the most confident choice among the four options. Note that for the gloss-level setting, only the video samples that occur in the WLASL val/test sets are counted, which amounts to 1,154 questions. For each setup, we report the results of models trained

Table 14: Evaluation results (%) on the subunit-level questions in LexSign-Bench. AR, PE, PA, and BO denote the results for arbitrary, perceptual, pantomimic, and combined perceptual–pantomimic glosses, respectively. AVG indicates the average results across different iconicity types.

| MLLMs | AVG | AR | PE | PA | BO |
|---|---|---|---|---|---|
| *Conventional Models* | | | | | |
| ZSSLR Model (full training set) | 51.3 | 46.9 | 53.1 | 54.4 | 50.8 |
| ZSSLR Model (2/3 training set) | 51.6 | 48.9 | 52.8 | 55.3 | 49.6 |
| ZSSLR Model (1/3 training set) | 51.0 | 49.1 | 52.2 | 52.8 | 50.0 |
| *Closed-source MLLMs* | | | | | |
| GPT-5 | 62.6 | 55.8 | 61.9 | 60.8 | 71.9 |
| Gemini 2.5 Pro | 58.0 | 52.0 | 60.3 | 59.7 | 59.8 |
| Qwen-VL-Max | 48.3 | 44.6 | 50.2 | 44.7 | 53.5 |
| *Open-source MLLMs* | | | | | |
| InternVL3.5-8B | 38.7 | 37.9 | 35.3 | 36.0 | 45.4 |
| Qwen2.5-VL-7B | 40.5 | 35.4 | 37.1 | 40.6 | 48.9 |
| LLaVA-OneVision-7B | 39.3 | 36.2 | 37.8 | 38.0 | 45.2 |

Table 15: Evaluation results (%) on the 1,154 gloss-level questions in LexSign-Bench. AR, PE, PA, and BO denote the results for arbitrary, perceptual, pantomimic, and combined perceptual–pantomimic glosses, respectively. AVG indicates the average results across all iconicity types.

| MLLMs | AVG | AR | PE | PA | BO |
|---|---|---|---|---|---|
| *Conventional Models* | | | | | |
| ISLR Model (full training set) | 72.1 | 70.5 | 69.4 | 73.6 | 75.0 |
| ISLR Model (2/3 training set) | 71.0 | 68.0 | 68.7 | 72.2 | 75.0 |
| ISLR Model (1/3 training set) | 65.9 | 61.9 | 64.6 | 69.8 | 67.0 |
| ISLR Model (1/6 training set) | 53.0 | 48.4 | 51.5 | 53.5 | 58.7 |
| *Closed-source MLLMs* | | | | | |
| GPT-5 | 65.7 | 56.2 | 63.6 | 70.1 | 72.6 |
| Gemini 2.5 Pro | 60.0 | 54.4 | 59.6 | 65.6 | 60.1 |
| Qwen-VL-Max | 39.9 | 27.0 | 31.0 | 49.7 | 52.1 |
| *Open-source MLLMs* | | | | | |
| InternVL3.5-8B | 34.3 | 27.0 | 28.3 | 36.8 | 45.1 |
| Qwen2.5-VL-7B | 34.4 | 25.6 | 23.6 | 44.4 | 44.1 |
| LLaVA-OneVision-7B | 35.5 | 22.8 | 33.0 | 47.6 | 38.5 |

on one-third, two-thirds, and the full amount of training data, in order to assess the capability of MLLM relative to conventional deep learning models trained with varying data sizes. The results are presented in Table 14 and Table 15.

Our results show that GPT-5 performs similarly to an ISLR model trained on one-third of WLASL training data for gloss-level questions, and it outperforms the ZSSLR model on subunit-level questions. Note that the ZSSLR model was trained without access to any of the 300 glosses included in LexSign-Bench. However, it had seen a large number of distractor options during training. As a result, ZSSLR is making predictions under a GZSL setting, which explains its suboptimal performance. In summary, comparisons with conventional models reveal that state-of-the-art MLLMs contain rich sign language knowledge, enabling their application in sign language understanding.

## A.10 DETAILS OF LEXSIGN-BENCH

**Detailed Result of LexSign-Bench.** The detailed evaluation results on LexSign-Bench are presented in Table 16.

**Hard-distractors Mining Strategy.** To construct a more challenging multiple-choice benchmark, we first generate the most confusable classes for the ground-truth glosses predicted by an ISLR model. The lexical descriptions or glosses of these confusable classes are then used as distractors in the multiple-choice options. We perform an ablation study on LLaVA-OneVision-7B to evaluate the effectiveness of this strategy, as summarized in Table 17. Incorporating the hard-distractor mining strategy leads to a substantial drop in evaluation scores, thereby validating its efficacy.

Table 16: Evaluation Results (%) of MLLMs on LexSign-Bench. AR, PE, PA, and BO represent results on arbitrary, perceptual, pantomimic, and combined perceptual-pantomimic sign glosses, respectively. AVG (S) and AVG (G) denote the average evaluation results at the subunit level and gloss level, respectively. AVG is the average of the evaluation results. It should be noted that the user study included only 20 participants, each responding to 32 questions.

| MLLMs | AVG | Subunit-level Perception | | | | | Gloss-level Recognition | | | | |
|---|---|---|---|---|---|---|---|---|---|---|---|
| | | AVG (S) | AR | PE | PA | BO | AVG (G) | AR | PE | PA | BO |
| *Human* | | | | | | | | | | | |
| Signers | 85.3 | 85.0 | 82.5 | 82.5 | 92.5 | 82.5 | 85.6 | 77.5 | 87.5 | 95.0 | 82.5 |
| Non-signers | 70.3 | 78.1 | 77.5 | 70.0 | 75.0 | 90.0 | 62.5 | 27.5 | 87.5 | 62.5 | 72.5 |
| *Closed-source MLLMs* | | | | | | | | | | | |
| GPT-5 | 65.0 | 62.6 | 55.8 | 61.9 | 60.8 | 71.9 | 67.4 | 59.9 | 65.9 | 69.3 | 74.2 |
| Gemini 2.5 Pro | 58.8 | 58.0 | 52.0 | 60.3 | 59.7 | 59.8 | 59.6 | 54.0 | 60.4 | 66.0 | 58.0 |
| Qwen-VL-Max | 43.7 | 48.3 | 44.6 | 50.2 | 44.7 | 53.5 | 39.2 | 27.7 | 30.5 | 47.8 | 50.7 |
| *Open-source MLLMs* | | | | | | | | | | | |
| InternVL3.5-8B | 36.1 | 38.7 | 37.9 | 35.3 | 36.0 | 45.4 | 33.5 | 27.6 | 26.0 | 39.9 | 40.6 |
| Qwen2.5-VL-7B | 37.3 | 40.5 | 35.4 | 37.1 | 40.6 | 48.9 | 34.1 | 24.0 | 25.1 | 44.4 | 42.8 |
| LLaVA-OneVision-7B | 36.8 | 39.3 | 36.2 | 37.8 | 38.0 | 45.2 | 34.3 | 22.7 | 33.9 | 45.3 | 35.3 |

Table 17: Ablation (%) on FPS, input frames, and the sampling strategy on LLaVA-OneVision. AVG (S) and AVG (G) denote the average evaluation results at the subunit level and gloss level, respectively. AVG is the average of the evaluation results.

| FPS | Frames | Sampling strategy | AVG | AVG (S) | AVG (G) |
|---|---|---|---|---|---|
| 10 | whole frm | Random | 42.7 | 47.1 | 38.2 |
| 10 | central 32 frm | Random | 43.4 | 49.2 | 37.7 |
| 5 | central 16 frm | Random | 46.8 | 52.9 | 40.7 |
| 10 | whole frm | Hard | 34.8 | 36.4 | 33.3 |
| 10 | central 32 frm | Hard | 36.0 | 38.5 | 33.5 |
| 5 | central 16 frm | Hard | 36.8 | 39.3 | 34.3 |

**Impact of FPS and Frames of Input Video.** We input the entire video at 10 FPS into closed-source MLLMs. In contrast, since most open-source MLLMs are trained to process videos as sequences of individual frames, we provide them with centrally cropped videos consisting of 16 frames sampled at 5 FPS. We conduct an ablation study on LLaVA-OneVision-7B to investigate the effects of input FPS and the number of frames on evaluation results. It can be observed that providing the MLLM with more input frames led to a decrease in the evaluation result, thereby validating the validity of our experimental setup on FPS and input frames.

**Impact of Prompt Complexity.** To further investigate the impact of prompt complexity on benchmark results, we evaluated three open-source MLLMs using two alternative prompts of differing complexity levels from the original prompt. Detailed prompts are provided in Supplementary Sect. A.14. For simple prompts, MLLMs are not provided with any additional information. For complex prompts, MLLMs are explicitly guided to focus on specific details, such as hand shapes. As shown in Table 18, the complexity of prompts has little effect on outcomes of the benchmark evaluations. Since the goal of LexSign-Bench is to guide model selection, we recommend using fixed prompts and reporting the average performance across the three prompts described above to minimize the influence of prompt engineering on the LexSign-Bench and ensure fairer comparisons.

## A.11 ABLATION ON HYPERPARAMETERS

**Ablation on Loss Weights.** We conduct an ablation study on the loss weights $w_{\text{FG},s}$, $w_{\text{FG},u}$ (in Equ. 5) and $w_{\text{HC}}$ (in Equ. 7) in the ZSSLR experiments on LexSign-ASL1000, as shown in Table 19.

**Ablation on Batch Size and Temperature.** We conduct an ablation study on the batch size $B$ and temperature $\tau$ in the ZSSLR experiments on LexSign-ASL1000, as shown in Table 20.

Table 18: Results (%) across different prompts for subunit-level and gloss-level questions. AVG denotes the overall average accuracy across the three prompts of varying complexity (simple, normal, and complex) under the two levels (subunit level and gloss level).

| MLLM | AVG | Subunit-level Perception | | | Gloss-level Recognition | | |
| --- | --- | --- | --- | --- | --- | --- | --- |
| | | Simple | Normal | Complex | Simple | Normal | Complex |
| InternVL3.5-8B | 36.6 | 43.0 | 38.7 | 38.0 | 34.2 | 33.5 | 32.0 |
| Qwen2.5-VL-7B | 38.5 | 44.2 | 40.5 | 41.2 | 33.3 | 34.1 | 37.4 |
| LLaVA-OneVision-7B | 36.3 | 42.3 | 39.3 | 35.6 | 32.5 | 34.3 | 33.4 |

Table 19: Ablation (%) on loss weights in the ZSSLR experiments on LexSign-ASL1000.

| $w_{FG,s}$ | $w_{FG,u}$ | $w_{HC}$ | Val | | Test | |
| --- | --- | --- | --- | --- | --- | --- |
| | | | Top-1 | Top-5 | Top-1 | Top-5 |
| 0.5 | 1.0 | 1.0 | 34.3 | 62.0 | 26.2 | 52.7 |
| 2.0 | 1.0 | 1.0 | 35.1 | 63.2 | 27.1 | 53.4 |
| 1.0 | 0.5 | 1.0 | 33.8 | 62.6 | 26.7 | 53.2 |
| 1.0 | 2.0 | 1.0 | 35.0 | 62.6 | 26.8 | 53.1 |
| 1.0 | 1.0 | 1.0 | **35.3** | **63.8** | 27.6 | 53.1 |
| 1.0 | 1.0 | 0.5 | 35.0 | 63.3 | 27.2 | 53.0 |
| 1.0 | 1.0 | 2.0 | 35.0 | 61.8 | 27.3 | 52.9 |

Table 20: Ablation (%) on batch size and temperature on LexSign-ASL1000.

| $B$ | $\tau$ | Training Loss | Val | | Test | |
| --- | --- | --- | --- | --- | --- | --- |
| | | | Top-1 | Top-5 | Top-1 | Top-5 |
| 12 | 0.01 | $\mathcal{L}_{Global}$ | 30.9 | 58.1 | 23.1 | 48.6 |
| 12 | 0.01 | $\mathcal{L}_{HALI}$ | **37.1** | **64.3** | 28.5 | 54.3 |
| 12 | 0.1 | $\mathcal{L}_{Global}$ | 17.7 | 42.2 | 13.8 | 33.9 |
| 12 | 0.1 | $\mathcal{L}_{HALI}$ | 30.7 | 57.9 | 24.2 | 48.7 |
| 12 | 0.03 | $\mathcal{L}_{Global}$ | 26.5 | 54.2 | 21.0 | 44.2 |
| 12 | 0.03 | $\mathcal{L}_{HALI}$ | 35.3 | 63.8 | 27.6 | 53.1 |
| 4 | 0.03 | $\mathcal{L}_{Global}$ | 24.6 | 53.8 | 19.8 | 44.1 |
| 4 | 0.03 | $\mathcal{L}_{HALI}$ | 31.0 | 60.1 | 23.3 | 49.7 |
| 32 | 0.03 | $\mathcal{L}_{Global}$ | 26.4 | 54.0 | 20.1 | 43.7 |
| 32 | 0.03 | $\mathcal{L}_{HALI}$ | 34.7 | 63.1 | 27.5 | 53.1 |

Table 21: Ablation (%) on the number of selected subunit-level features on LexSign-ASL1000.

| Training Loss | $M$ | Val | | Test | |
| --- | --- | --- | --- | --- | --- |
| | | Top-1 | Top-5 | Top-1 | Top-5 |
| $\mathcal{L}_{HALI}$ | 2 | 33.3 | 61.0 | 26.8 | 52.5 |
| $\mathcal{L}_{HALI}$ | 3 | **35.3** | **63.8** | 27.6 | 53.1 |
| $\mathcal{L}_{HALI}$ | 5 | 34.6 | 62.4 | 26.6 | 52.9 |

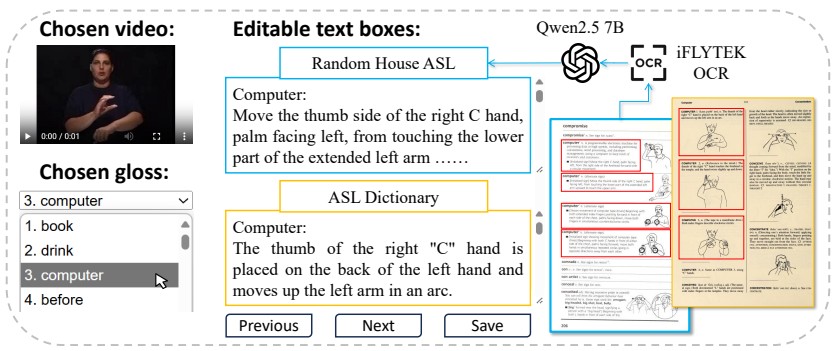

Figure 10: Illustration of the user interface of the designed annotation tool.

**Ablation on the Number of Selected Subunit-level Features.** We conduct an ablation study on the number of selected subunit-level features $M$ (in Equ. 8) in the ZSSLR experiments on LexSign-ASL1000, as shown in Table 21. For $M = 2$, the selected features are the left hand and right hand. For $M = 3$, the selected features are the body, left hand, and right hand. For $M = 5$, the selected features are the body, left hand, right hand, mouth, and face.

### A.12 DEMONSTRATION OF THE INTERFACE OF THE DESIGNED ANNOTATION TOOL

An illustration of the user interface of our designed annotation tool in MCP is shown in Fig. 10.

### A.13 LLM USAGE STATEMENT

In this paper, we use LLMs solely for text refinement and for generating the icons shown in Fig. 4 to address copyright considerations.

### A.14 PROMPTS FOR LEXSIGN-BENCH AND AGP

The original prompt for subunit-level evaluation in the LexSign-Bench is as follows:

> *In the middle of the input video, a person is performing American Sign Language (ASL). Each of the following options describes how a sign is performed, possibly using fingerspelled letters from American Fingerspelled Alphabet to indicate handshapes (e.g., A hand, F hand). Please select the option that best corresponds to the sign being performed:*
> *A) [Option A];*
> *B) [Option B];*
> *C) [Option C];*
> *D) [Option D].*
> *Respond only with the selected option: A, B, C or D.*

The *simple* prompt for subunit-level evaluation in the LexSign-Bench is as follows:

> *Describe the sign performed in the video from the following options:*
> *A) [Option A];*
> *B) [Option B];*
> *C) [Option C];*
> *D) [Option D].*
> *Respond only with the selected option: A, B, C or D.*

The *complex* prompt for subunit-level evaluation in the LexSign-Bench is as follows:

> *You are tasked with analyzing a video segment where an individual is using American Sign Language (ASL). Your primary goal is to meticulously describe the production of a specific sign. To ensure a comprehensive and accurate analysis, you must consider all five fundamental parameters of ASL: handshape, palm orientation, location, movement, and non-manual markers. For clarity, you may use fingerspelled letters from the American Fingerspelled Alphabet to specify handshapes (e.g., A hand, F hand).*
> *Your description should be detailed and structured, addressing each of the following components:*
> *1. Handshape: Describe the specific shape of the hand or hands used to form the sign. Note if the handshape corresponds to a letter from the American Fingerspelled Alphabet (e.g., a "C" handshape, a "5" handshape). Detail the configuration of the fingers and thumb. For instance, are the fingers extended, bent, or closed? Is the thumb tucked or extended?*
> *2. Palm Orientation: Specify the direction the palm is facing. Is it oriented upwards, downwards, forwards (away from the signer), backwards (towards the*

*signer), or to the side? If the orientation changes during the execution of the sign, describe this change.*

*3. Location: Identify the location on or near the body where the sign is produced. This could be in front of the chest, near the forehead, on the chin, or in neutral space in front of the signer. Be precise about the starting and ending locations if the sign involves movement between two points.*

*4. Movement: Detail the action of the hand or hands. Is the movement a straight line, a circular motion, a tapping motion, or a wrist twist? Describe the directionality of the movement (e.g., upward, downward, forward, side-to-side). If there are repeated movements, specify the number of repetitions.*

*Please select the option that best corresponds to the sign being performed:*

*A) [Option A];*

*B) [Option B];*

*C) [Option C];*

*D) [Option D].*

*Respond only with the selected option: A, B, C or D.*

The original prompt for gloss-level evaluation in the LexSign-Bench is as follows:

*In the middle of the input video, a person is performing American Sign Language (ASL). Choose the sign being performed from the following options:*

*A) [Option A];*

*B) [Option B];*

*C) [Option C];*

*D) [Option D].*

*Respond only with the selected option: A, B, C or D.*

The *simple* prompt for gloss-level evaluation in the LexSign-Bench is as follows:

*Identify the sign in the video from the following options:*

*A) [Option A];*

*B) [Option B];*

*C) [Option C];*

*D) [Option D].*

*Respond only with the selected option: A, B, C or D.*

The *complex* prompt for gloss-level evaluation in the LexSign-Bench is as follows:

*Your task is to perform a precise visual analysis of the provided input video. You must focus your attention specifically on the temporal midpoint of the clip. In this middle section of the video, an individual is demonstrating a specific sign from American Sign Language (ASL).*

*Your objective is to accurately identify this sign. Carefully observe the performer's handshape, palm orientation, location, and the specific movement of the sign. Compare these visual components against the list of candidate options provided below.*

*Choose the sign being performed from the following options:*

*A) [Option A];*

*B) [Option B];*

*C) [Option C];*

*D) [Option D].*

*Respond only with the selected option: A, B, C or D.*

The prompt for generating lexical description for MLLMs is as follows:

*You are a linguist and lexicographer with expertise in American Sign Language (ASL). Your task is to analyze an ASL sign from a video and compose an authoritative, dictionary-style description.*

*[Objective]*

*Analyze the provided video of a single American Sign Language (ASL) sign and generate a formal, descriptive entry suitable for an authoritative sign language dictionary. The description must be a single, elegant paragraph that precisely integrates the sign's core linguistic parameters: Handshape, Palm Orientation, Location, and Movement.*

*[Output Requirements]*

*Your description must be a single, polished paragraph that cohesively integrates the four core articulatory parameters:*

*1. Handshape: The form of the dominant and non-dominant hands (e.g., open hand, 5 hand, 10 hand, A hand, B hand, F hand, I hand, L hand, V hand, Y hand, curved 5 hand, modified X hand).*

*2. Location: The position of the hands in signing space (e.g., in front of the chest, at the chin, at the temple).*

*3. Palm Orientation: The direction the palm faces (e.g., palm facing in, palm facing out, palm facing left, palm facing right, palm facing up, palm facing down).*

*4. Movement: The path and quality of the action.*

*Maintain a rigorous, objective, and clinical tone. Use precise, non-colloquial language and avoid metaphors. Do not describe handshapes using terms such as "5-hand", "'A' hand", "hand in a B handshape", or "hand in an 'A' handshape". As shown in the examples in the [Output Requirements], use "A hand", "open hand", etc., to describe the handshape, and use "palm facing in", etc., to describe the palm orientation. For example: "A hand, palm facing in". Follow the style, structure, and level of detail found in the [Examples]. Directly output the description of the sign, not to exceed 50 words, without any additional preamble or explanation.*

*[Examples]*

*1. [Example 1].*

*2. [Example 2].*

*3. [Example 3].*

The prompt for summarizing lexical description for LLMs is as follows:

*You are a linguist and lexicographer with expertise in American Sign Language (ASL). Your task is to synthesize multiple written descriptions of a single ASL sign into one authoritative definition for an official dictionary. This definition must be clear, precise, and serve as the standard for learners.*

*[Process]*

*1. Analyze Core Components: Begin by carefully reading and comparing all provided descriptions. For each description, you must identify the four core components of the sign: Handshape, Palm Orientation, Location, and Movement.*

*2. Synthesize Common Features: Distinguish the core, consistent features of the sign from any individual variations. Compare the components identified in the previous step and isolate the recurring patterns that are present across the vast majority of descriptions. You must discard details that are unique to a single source, as these likely represent personal signing habits, pauses, or recording artifacts rather than the sign's essential structure.*

*3. Construct the Final Description: Using the synthesized common features, generate the final dictionary entry. The description must be crafted with precise, professional terminology, yet remain simple and clear enough for a beginner to understand and accurately replicate the sign. Maintain an objective, rigorous, and formal tone suitable for an authoritative dictionary entry, avoiding any colloquialisms or metaphors.*

*4. Output the Final Description: Deliver the dictionary entry directly, without any introductory or concluding phrases. Do not describe handshapes using terms such as "5-hand", "'A' hand", "hand in a B handshape", or "hand in an 'A' handshape". Use "A hand", "open hand", etc., to describe the handshape, and use "palm facing in", etc., to describe the palm orientation. For example: "A hand, palm facing in". The entry must be under 50 words and adhere to the style, structure, and level of detail found in the [Examples].*

*[Examples]*

*1. [Example 1].*

*2. [Example 2].*

*3. [Example 3].*

*Original Sign Descriptions (from multiple videos):*

*Description 1: [Description 1]*

*Description 2: [Description 2]*

*Description 3: [Description 3]*

