# OpenReview forum: "LexSign: Learning Sign Language from Lexical Descriptions"
_ICLR.cc/2026/Conference — Submitted to ICLR 2026_

### Official Review · Reviewer_hpXN · 2025-10-26

**Soundness:** 3
**Presentation:** 2
**Contribution:** 3
**Rating:** 4
**Confidence:** 4

**Summary:**

This paper presents LexSign, a large-scale dataset of lexical descriptions for sign language signs, collected both manually from authoritative sign language dictionaries and automatically via a “perceive-then-summarize” pipeline using multimodal large language models (MLLMs) and LLMs. To ensure the quality of generated descriptions, the authors introduce LexSign-Bench, a comprehensive benchmark evaluating MLLMs’ ability to understand sign language at subunit, gloss, and sentence levels. Based on LexSign, they propose HALI (Hierarchical Action-Language Interaction), a novel framework that aligns sign language videos with lexical descriptions across multiple granularities—video, snippet, subunit, and textual fragments—through multi-granularity contrastive learning and hierarchical consistency constraints. Experiments on public datasets (e.g., ASL-Text, WLASL) demonstrate that HALI significantly improves performance in both zero-shot and supervised isolated sign language recognition, highlighting the value of fine-grained linguistic structure in advancing sign language understanding.

**Strengths:**

1. The problem proposed in this paper carries significant real-world relevance, and the LexSign dataset can greatly facilitate unified understanding and generation of sign language.
2. The approach that combines contrastive learning with hierarchical feature alignment is relatively novel.
3. The paper presents fairly comprehensive experiments that effectively support its claims.

**Weaknesses:**

1. The paper suffers from substantial writing issues, particularly concerning focus and the missing of details.
2. Although the proposed hierarchical consistency loss yields strong empirical results, the paper lacks a clear motivation or justification for its design.

**Questions:**

1. Writing issues:
a) The paper devotes excessive space to the method section. I recommend rebalancing the narrative in a revision by strengthening the description of dataset construction, for instance, by moving Figure 6 and its accompanying explanation from the supplementary material into the main paper.
b) The experimental section omits critical implementation details: What model serves as the baseline? What training protocol is used? How do the experimental setups differ across datasets?

2.  Hierarchical consistency concerns:
a) The definitions of snippet-level and subunit-level representations are ambiguous. Are they meaningfully distinct? If the granularity is merely defined by manual segmentation without linguistic grounding, does it carry genuine semantic significance?
b) Do Equations 5 and 8 include weighting hyperparameters to balance the contributions of different loss terms? Moreover, the paper lacks sensitivity analyses for key hyperparameters such as M , batch size B , and temperature τ .

3. Additional suggestions:
a) The set of baseline methods appears outdated; the authors should include comparisons with more recent state-of-the-art approaches.
b) Consider integrating the MLLM evaluation results from Table 10 into the main paper, and, if feasible, add a direct comparison between MLLM-based and traditional methods.
c) Could HALI be extended to MLLM SFT? Such an extension would significantly strengthen the paper’s impact and practical relevance.

---

> ### Author Response · Authors · 2025-11-24
> **Official Comment by Authors (1/5)**
>
> We appreciate the reviewers’ careful evaluation and insightful comments, which have helped us clarify and strengthen our work.
>
> > [W1] about writing issues
>
> Thank you for your helpful feedback. We have rebalanced and refined the narrative and added more important details. The primary modifications are summarized as follows.
>
> 1. The description of the data construction process has been strengthened. We have rebalanced the narrative by moving Figure 6 into the main paper.
>
> 2. Extensive experimental details (including the baseline model selection, the training protocol, hyperparameters, and more) have been added to the Implementation Details section of the main paper and Appendix A.1. We have made every effort to ensure full reproducibility of our experiments and to address the reviewers’ concerns regarding the experimental setup.
>
> 3. Key concepts, such as the definitions of lexical descriptions and fine-grained representations, are placed in prominent locations to help readers locate and understand them quickly.
>
> 4. Abbreviations and symbols have been made consistent throughout the paper, defined upon first appearance, and annotated in the headers of tables or captions of figures where they are used.
>
> > [W2] about motivation and justification for hierarchical consistency loss
>
> The original motivation of this work is how to improve the generalization ability of sign language translation models. Recent works [r1, r2] have attempted to use **automatically generated** sign language descriptions to enhance sign language translation (L162–164). **However, the performance gap between manually curated linguistic definitions and automatically generated descriptions remains insufficiently understood.** The proposed LexSign-Bench provides a valuable benchmark for evaluating MLLMs’ sign language understanding capabilities, while domain-specific MLLMs can be assessed by the quality of their generated captions on LexSign-ASL/CSL1000 in zero-shot and generalized zero-shot learning (ZSL/GZSL) settings.
>
> Considering the hierarchical structure of sign language, we further propose a HALI framework, which builds more accurate snippet-level and subunit-level alignments with multi-granularity contrastive loss and incorporates a consistency constraint between visual representations at different levels to leverage the inherent hierarchical structure of sign language. This approach enables fine-grained representations, supports potential translation across dialects and popular slang, and facilitates detailed motion generation [r3].
>
> Reference
> - [r1] Asasi S, Lakhal M I, Sincan O M, et al. Beyond Gloss: A Hand-Centric Framework for Gloss-Free Sign Language Translation[J]. arXiv preprint arXiv:2507.23575, 2025.
> - [r2] Kim J, Jeon H, Bae J, et al. Leveraging the power of mllms for gloss-free sign language translation[C]//Proceedings of the IEEE/CVF International Conference on Computer Vision. 2025: 21048-21058.
> - [r3] Bensabath L, Petrovich M, Varol G. Text-Driven 3D Hand Motion Generation from Sign Language Data[J]. arXiv preprint arXiv:2508.15902, 2025.
>
> > [Q1.a] about narrative rebalancing
>
> We thank the reviewer for the valuable suggestion. We have added more details on how the lexical descriptions are generated and included Figure 6 in the main paper. For our baseline model, we adopt CoSign-1s as the default visual encoder due to its robustness, and BERT as the default textual encoder. Models trained solely with $L_{\text{Global}}$ are the baseline for the ZSSLR task, while those trained solely with cross-entropy loss are the baseline for the ISLR task. Experimental setups across datasets maintain consistent (details are provided in Appendix A.1). We have expanded the Implementation Details section of the revised paper to ensure full reproducibility of our experiments.

---

> ### Author Response · Authors · 2025-11-24
> **Official Comment by Authors (2/5)**
>
> > [Q2.a] about definitions and the semantic significance of fine-grained representations
>
> Sign language conveys meaning through the sequential and simultaneous composition of phonological subunits [r1]. Lexical descriptions likewise consist of fragments that capture fine-grained cues (e.g., handshape, movement, orientation). To model this hierarchy, we represent textual representation at both the sentence and fragment levels. Similarly, we decompose each sign language video into video-level and snippet-level visual representations to maintain consistent hierarchical alignment.
>
> However, in our experiments, we found that aligning fragment-level and snippet-level representations is prone to inducing spurious correlations. For instance, the description “touch the extended right index finger to the right side of the nose, palm facing down” may cause the model to focus only on the relationship between the right index finger and the nose, while ignoring the overall posture of the fingers (e.g., whether the index or middle finger is used). Therefore, we further introduce the subunit-level features to capture fine-grained information.
>
> Specifically, a **snippet-level representation** is the feature extracted from *a whole-body sequence clip that potentially corresponds to a fragment within a lexical description*. It is implemented via pyramidal aggregation of the 1/2- and 1/4-scale frame features, yielding a snippet-level representation of length 6 (2+4) for each isolated sign. Similarly, a **subunit-level representation** is the feature extracted from *a part-level sequence clip that potentially corresponds to a fragment within a lexical description*, with the implementation identical to the snippet-level representation except for the input sequence. The Implementation Details contains a thorough description of how the fine-grained representations are generated.
>
> Both snippet-level and subunit-level representations are guided by the same semantic supervision. However, they are expected to capture different kinds of semantic information (global hand-body posture vs. detailed finger movement, as demonstrated in the above example). Moreover, the decomposition inherent in subunit-level representations provides more precise linguistic grounding, while snippet-level representations help mitigate noisy correspondences. As shown in Table 7, combining both levels leads to significant improvements, highlighting their complementary roles.
>
> | (s,f) | (u,f) | $L_{HC}$ | Val T-1 | Val T-5 | Test T-1 | Test T-5 |
> | :---: | :---: | :---: | :---: | :---: | :---: | :---: |
> |  |    |   | 26.5  | 54.2  | 21.0  | 44.2  |
> | ✓   |   | | 32.4  | 59.8  | 24.6  | 49.7  |
> |   | ✓ | | 29.7  | 57.1  | 22.8  | 48.5  |
> | ✓  | ✓  |  | 34.2  | 61.1  | 25.7  | 51.3  |
> | ✓   | ✓  | ✓  | **35.3**  | **63.8**  | 27.6  | 53.1  |
>
> We also visualize the distribution of description fragments in Fig.9, providing further insight into the significance of modeling subunit-level representations.
>
> [r1] Sandler W, Lillo-Martin D C. Sign language and linguistic universals[M]. Cambridge University Press, 2006.

---

> ### Author Response · Authors · 2025-11-24
> **Official Comment by Authors (3/5)**
>
> > [Q2.b] about weighting hyperparameters and sensitivity analyses for key hyperparameters
>
> In the original implementation, the weights of all loss terms ($w_{Global}$, $w_{\text{FG},s}$, $w_{\text{FG},u}$, and $w_{\text{HC}}$) are balanced. We have incorporated weighting hyperparameters to adjust the contributions of the different loss terms in Equations 5 and 8. Regarding hyperparameter tuning for the proposed method, we provide ablation results on loss weights, batch size, and the number of selected subunit-level features. Our analysis shows that reducing the temperature in contrastive learning from 0.03 to 0.01 leads to a notable performance gain, improving the validation Top-1 accuracy from 35.3% to 37.1% on LexSign-ASL1000. We sincerely thank your valuable suggestion and will update all experimental results in the next version. A more detailed hyperparameter ablation study can be found in Appendix A.11.
>
> *Ablation(%) on loss weights in the ZSSLR experiments on LexSign-ASL1000.*
>
> | $w_{\text{FG},s}$ | $w_{\text{FG},u}$ | $w_{\text{HC}}$ | Val Top-1 | Val Top-5 | Test Top-1 | Test Top-5 |
> |:---:|:---:|:---:|:---:|:---:|:---:|:---:|
> | 0.5 | 1.0 | 1.0 | 34.3 | 62.0 | 26.2 | 52.7 |
> | 2.0 | 1.0 | 1.0 | 35.1 | 63.2 | 27.1 | 53.4 |
> | 1.0 | 0.5 | 1.0 | 33.8 | 62.6 | 26.7 | 53.2 |
> | 1.0 | 2.0 | 1.0 | 35.0 | 62.6 | 26.8 | 53.1 |
> | 1.0 | 1.0 | 1.0 | **35.3** | **63.8** | 27.6 | 53.1 |
> | 1.0 | 1.0 | 0.5 | 35.0 | 63.3 | 27.2 | 53.0 |
> | 1.0 | 1.0 | 2.0 | 35.0 | 61.8 | 27.3 | 52.9 |
>
> *Ablation(%) on batch size and temperature on LexSign-ASL1000.*
>
> | $B$ | $\tau$ | Training Loss | Val Top-1 | Val Top-5 | Test Top-1 | Test Top-5 |
> |:---:|:------:|:-------------:|:---------:|:---------:|:----------:|:----------:|
> | 12 | 0.01 | $\mathcal{L}_{\text{Global}}$ | 30.9  | 58.1 | 23.1 | 48.6  |
> | 12 | 0.01 | $\mathcal{L}_{\text{HALI}}$   | **37.1**| **64.3**  | 28.5  | 54.3 |
> | 12 | 0.1| $\mathcal{L}_{\text{Global}}$ | 17.7  | 42.2  | 13.8 | 33.9 |
> | 12 | 0.1 | $\mathcal{L}_{\text{HALI}}$   | 30.7 | 57.9  | 24.2   | 48.7 |
> | 12 | 0.03 | $\mathcal{L}_{\text{Global}}$ | 26.5   | 54.2 | 21.0  | 44.2  |
> | 12 | 0.03 | $\mathcal{L}_{\text{HALI}}$   | 35.3   | 63.8 | 27.6   | 53.1  |
> | 4 | 0.03 | $\mathcal{L}_{\text{Global}}$ | 24.6   | 53.8 | 19.8    | 44.1  |
> | 4 | 0.03 | $\mathcal{L}_{\text{HALI}}$   | 31.0 | 60.1 | 23.3    | 49.7   |
> | 32  | 0.03 | $\mathcal{L}_{\text{Global}}$ | 26.4   | 54.0 | 20.1   | 43.7 |
> | 32  | 0.03   | $\mathcal{L}_{\text{HALI}}$   | 34.7   | 63.1 | 27.5    | 53.1    |
>
> *Ablation(%) on the number of selected subunit-level features on LexSign-ASL1000.*
>
> | Training Loss | $M$ | Val Top-1 | Val Top-5 | Test Top-1 | Test Top-5 |
> |:-------------:|:---:|:---------:|:---------:|:----------:|:----------:|
> | $\mathcal{L}_{\text{HALI}}$ | 2  |   33.3  |  61.0   |   26.8    |  52.5    |
> | $\mathcal{L}_{\text{HALI}}$ | 3 | **35.3**  | **63.8**  | 27.6   | 53.1 |
> | $\mathcal{L}_{\text{HALI}}$ | 5  | 34.6   | 62.4      | 26.6       | 52.9   |

---

> ### Author Response · Authors · 2025-11-24
> **Official Comment by Authors (4/5)**
>
> > [Q3.a] about comparisons with more recent state-of-the-art approaches
>
> Thank you for the suggestion of Reviewer hpXN. To verify the effectiveness of the proposed method, we reproduce two recently proposed zero-shot skeleton-based action recognition methods [r2,r3] on the ASL-Text dataset as new comparison baselines and include the results in Table 1. As shown in the table below, the proposed HALI method consistently outperforms these recent baselines across different settings.
>
> | Method | ZSL T-1 | ZSL T-5 | GZSL-S T-1 | GZSL-S T-5 | GZSL-U T-1 | GZSL-U T-5 | GZSL-H T-1 | GZSL-H T-5 |
> | :--- | :---: | :---: | :---: | :---: | :---: | :---: | :---: | :---: |
> | LLE [r1]  | 24.7 | 54.6 | **81.9** | 95.5 | 2.1 | 18.4 | 4.1 | 30.7 |
> | DVTA [r2] | 17.0 | 47.2 | 19.6 | 61.0 | 2.6 | 12.4 | 4.6 | 20.4 |
> | PGFA [r3] | 26.8 | 51.9 | 62.2 | 86.9 | 2.1 | 11.2 | 4.1 | 19.9 |
> | Baseline | 29.9 | 68.3 | 76.3 | 95.1 | 5.4 | 31.0 | 10.1 | 46.7 |
> | **Ours** | **40.1** | **74.3** | 81.1 | **95.8** | **6.4** | **33.0** | **11.9** | **49.0** |
>
> Moreover, as illustrated in the table below and in Table 11, we conduct zero-shot sign language recognition experiments on both LexSign-ASL/CSL1000 to further validate the generalizability of our proposed HALI. It is observed that incorporating the proposed fine-grained loss and hierarchical consistency loss with DVTA and PGFA consistently and significantly boosts performance, confirming the generalization ability of HALI.
>
> | Model | $L_{\text{FG}}$ + $L_{\text{HC}}$ | Val T-1 (ASL) | Val T-5 (ASL) | Test T-1 (ASL) | Test T-5 (ASL) | Val T-1 (CSL) | Val T-5 (CSL) | Test T-1 (CSL) | Test T-5 (CSL) |
> | :---: | :---: | :---: | :---: | :---: | :---: | :---: | :---: | :---: | :---: |
> | DVTA [r2] | | 21.3 | 46.8 | 16.1 | 37.6 | 16.8 | 43.2 | 10.8 | 33.5 |
> | DVTA [r2] | $\checkmark$ | 32.5 | 60.8 | 25.2 | 51.0 | 37.9 | 65.8 | 29.1 | 56.8 |
> | PGFA [r3] | | 32.1 | 53.3 | 25.1 | 43.2 | 39.4 | 59.1 | 29.8 | 48.3 |
> | PGFA [r3] | $\checkmark$ | 34.8 | 56.2 | **28.1** | 46.8 | 42.0 | 61.7 | 31.9 | 50.6 |
> | Baseline | | 26.5 |  54.2 |  21.0 |  44.2 | 39.9 | 68.7 | 30.7 | 59.7 |
> | Ours | $\checkmark$ | **35.3** |  **63.8** | 27.6 | **53.1** | **44.9** |  **72.6** | **36.2** | **62.0** |
>
> Reference
> - [r1] Bilge Y C, Cinbis R G, Ikizler-Cinbis N. Towards zero-shot sign language recognition[J]. IEEE Transactions on Pattern Analysis and Machine Intelligence, 2022, 45(1): 1217-1232.
> - [r2] Kuang J, Wang H, Han C, et al. Zero-shot skeleton-based action recognition with dual visual-text alignment[J]. Pattern Recognition, 2025: 112342.
> - [r3] Zhou K, Zhang S, You Z, et al. Zero-shot skeleton-based action recognition with prototype-guided feature alignment[J]. in IEEE Transactions on Image Processing, vol. 34, pp. 4602-4617, 2025, doi: 10.1109/TIP.2025.3586487.

---

> ### Author Response · Authors · 2025-11-24
> **Official Comment by Authors (5/5)**
>
> > [Q3.b] about integrating Table 10 into the main paper
>
> We sincerely thank reviewer hpXN for the helpful suggestion to integrate the MLLM evaluation results from Table 10 into the main paper. Since Table 10 reports the same results shown in Figure 4, we place Figure 4 in the main paper and retain Table 10 in the supplementary material for completeness, freeing space in the main paper for other content.
>
> > [Q3.b] about the comparison between MLLM-based and traditional methods
>
> To provide a more thorough evaluation of MLLMs’ performance on LexSign-Bench, we conducted both a user study and a comparative analysis with conventional deep learning models. In the user study, we enlisted 10 signers and 10 non-signers, each of whom completed a 32-question survey spanning two difficulty levels and four iconicity types. This enabled us to collect human performance data from both groups (signers and non-signers) for direct comparison. For the conventional baselines, we used our proposed ZSSLR model for subunit-level evaluation and the ISLR model for gloss-level evaluation. Both models were trained on the LexSign dataset **with no benchmark data overlap**. The results are summarized below, with additional details provided in the main paper and in Appendix A.9.
>
> | MLLMs | AVG (S) | AVG (G) | &nbsp;&nbsp; | Settings | AVG (S) | AVG (G) |
> | :--- | :---: | :---: | :---: | :--- | :---: | :---: |
> | *Closed-source MLLMs* | | | | *Human* | | |
> | GPT-5 | 62.6 | 65.7 | | Signer | 85.0 | 85.6 |
> | Gemini 2.5 Pro | 58.0 | 60.0 | | Non-signer | 78.1 | 62.5 |
> | Qwen-VL-Max | 48.3 | 39.9 | | *Conventional Models* | | |
> | *Open-source MLLMs* | | | | ZSSLR Model (full training set) | 51.3 | - |
> | InternVL3.5-8B | 38.7 | 34.3 | | ISLR Model (full training set) | - | 72.1 |
> | Qwen2.5-VL-7B | 40.5 | 34.4 | | ISLR Model (2/3 training set) | - | 71.0 |
> | LLaVA-OneVision-7B | 39.3 | 35.5 | | ISLR Model (1/3 training set) | - | 65.9 |
> | | | | | ISLR Model (1/6 training set) | - | 53.0 |
>
> In summary, GPT-5 demonstrates strong performance on gloss-level questions, achieving an accuracy of 65.7%, which is comparable to the ISLR model trained on one-third of the WLASL dataset (65.9%). Moreover, GPT-5 outperforms the ZSSLR model on subunit-level questions, indicating superior generalization capability. Notably, GPT-5 even exceeds the performance of non-signers on gloss-level tasks (65.7% vs. 62.5%), highlighting its impressive aptitude for sign language understanding. Collectively, these comparisons indicate that state-of-the-art MLLMs possess substantial sign language understanding ability (under four-choice questions). However, as illustrated in Tables 2 and 5, they still fall short of generating consistently reliable descriptions.
>
> > [Q3.c] about extending HALI to MLLM SFT
>
> We thank Reviewer hpXN for the suggestion. Recent works [r1, r2] have also attempted to use **automatically generated** sign language descriptions to enhance sign language translation (L162–164). **However, the performance gap between manually curated linguistic definitions and automatically generated descriptions remains insufficiently understood.** The proposed LexSign-Bench provides a valuable benchmark to evaluate MLLMs’ sign language understanding capabilities, while the proposed HALI framework can further improve the utilization of lexical descriptions for downstream tasks such as sign language translation. Given the challenges of constructing a dataset that contains both high-quality lexical descriptions and aligned sign language translation data, we leave this as a direction for future work.
>
> - [r1] Asasi S, Lakhal M I, Sincan O M, et al. Beyond Gloss: A Hand-Centric Framework for Gloss-Free Sign Language Translation[J]. arXiv preprint arXiv:2507.23575, 2025.
> - [r2] Kim J, Jeon H, Bae J, et al. Leveraging the power of mllms for gloss-free sign language translation[C]//Proceedings of the IEEE/CVF International Conference on Computer Vision. 2025: 21048-21058.

---

### Official Review · Reviewer_WGdq · 2025-10-28

**Soundness:** 3
**Presentation:** 3
**Contribution:** 3
**Rating:** 6
**Confidence:** 4

**Summary:**

The paper focuses on sign language recognition from the perspective of fine-grained recognition with vision-language models. For this, paper has three main contributions: (i) a new dataset (and a new benchmark), (ii) a training loss structure that aims to enforce learning vision-language alignment for sign language recognition in a hierarchical and fine-grained manner, and (iii) a comprehensive evaluation that involves existing open+closed source multimodal LLMs.


The dataset is collected in a semi-manual manner. Human annotators matched videos with descriptions at lexical description level. In addition, a combination of LLM+MLLM based heuristic is used to add lexical descriptions to other videos.

The proposed method relies on an enhanced CLIP-like training mechanism. The core method relies on contrastive aligment loss across the sign videos and descriptions. This is enhanced by introducing additional loss terms based on (i) video-snippet & description-fragments (eq 4) and (ii) a term enforcing consistency across the similarity distributions over fragments and video snippets versus fragments and subunits.

**Strengths:**

The makes contributions on three valuable ends: (i) sign language datasets & benchmarks, (ii) hierarchical vision-language formulations for sign video understanding and (iii) evaluation of MLLM abilities.

The paper is overall well written (though I do have several confusions, as listed below in the questions field).

The proposed method is sensible.

The experimental results contain several interesting bits.

Commitment to make the data & techniques publicly available that ensures full reproducibility.

**Weaknesses:**

Some important details are hard to find in the paper, and some of the experimental result discussions are confusing (see below).

There is no clear model selection protocol defined in the paper for the proposed benchmark, which may incorrectly encourage model selection directly over the test results -- a huge problem for any (future) researcher who wants to do it right and avoid any test-set based hyper-parameter tuning. For instance, one can imagine an extremely complex prompt that simply maximizes the MLLM performance on the proposed test set, w/o actually contributing to the real-world generalization.

No discussion on model selection & hyper-parameter tuning for the proposed method.

**Questions:**

Can please summarize the techniques to split video & text data into subunits, video snippets and description fragments with points to the detailed descriptions in the paper? I understand the concepts but also curious about how to practically obtain them, currently I am bit lost here.


Are the comparisons to prior work in Table 1 & Table 2 fair? Do you use the same training data with those models in comparison?


I understand you measure the quality of auto-generated descriptions by using them for training and evaluating on the manually collected dataset. However, I think here there is a missing baseline that involves training with manually collected data. This evaluation could have been more complete if there was a version that used a different data split where some of the manually annotated data is used at training time (and comparing against use those images with auto-generated descriptions).

What is the conclusion on the performance of general purpose MLLMs versus training ad-hoc sign models?

**Details Of Ethics Concerns:**

-

---

> ### Author Response · Authors · 2025-11-24
> **Official Comment by Authors (1/4)**
>
> We sincerely appreciate the reviewers’ efforts in evaluating our paper. We hope that the following responses address all concerns clearly and satisfactorily.
>
> > [W1] about important details and experimental result discussions
>
> Thank you for your valuable feedback. We have added the important details and revised the discussion of the experimental results accordingly. The primary modifications are summarized as follows.
>
> 1. Extensive experimental details (including the model selection protocol, hyperparameters, training configuration, and more) have been added to the Implementation Details section of the main paper, the Method section, and Appendix A.1. We have made every effort to ensure full reproducibility of our experiments and to address the reviewers’ concerns regarding the experimental setup.
>
> 2. Discussions of the experimental results have been further elaborated. We added statements addressing the fairness of comparisons among different models across different tasks. Additionally, we reported results from conventional deep learning models and conducted a user study on LexSign-Bench, accompanied by discussions that further support the validity of our findings.
>
> Moreover, we have incorporated the following changes to enhance the clarity and readability of the manuscript.
>
> 1. Abbreviations and symbols have been made consistent throughout the paper, defined upon first appearance, and annotated in the headers of tables or captions of figures where they are used.
>
> 2. Key concepts, such as the definitions of lexical descriptions and fine-grained representations, are placed in prominent locations to help readers locate and understand them quickly.
>
> 3. The figure depicting the proposed Manual Curation Pipeline (MCP) and Automated Generation Pipeline (AGP) has been relocated to the main paper to improve clarity. We believe that the figure makes it easier to grasp AGP and MCP.

---

> ### Author Response · Authors · 2025-11-24
> **Official Comment by Authors (2/4)**
>
> > [W2] about model selection protocol
>
> We thank the reviewer for raising this important point regarding the potential risk of performing model selection directly on the test set. The evaluation of MLLMs on LexSign can be divided into two settings:
>
> 1. **Understanding the sign language understanding ability of MLLMs.** As shown in the paper, we select three advanced closed-source MLLMs and three open-source 7B/8B MLLMs, and all MLLMs are evaluated with **the same prompt**. We deliberately used very concise prompts for MLLMs, which are detailed below (denoted as *Normal*) and in Appendix A.14. For gloss-level questions, MLLMs are only informed that the video is in American Sign Language. For subunit-level questions, MLLMs are additionally told that handshapes may be described using the American Fingerspelled Alphabet. These extremely concise prompts ensure that no extra information is leaked to the MLLMs, guaranteeing the generalizability of the results.
>
>     To further investigate the impact of prompt complexity on benchmark results, we evaluated three open-source MLLMs using two alternative prompts of differing complexity levels from the original prompt, referred to here as *Simple* and *Complex*. For simple prompts, MLLMs are not provided with any additional information. For complex prompts, MLLMs are explicitly guided to focus on specific details, such as hand shapes.
>
>     As shown in the table, the complexity of prompts has little effect on outcomes of the benchmark evaluations. Since the goal of LexSign-Bench is to guide model selection, we recommend **using fixed prompts and reporting the average performance across the three prompts described above** to minimize the influence of prompt engineering on the LexSign-Bench and ensure fairer comparisons. The updated results are provided in Table 18.
>
> | MLLM                 | AVG  | AVG (S) | Simple (S) | Normal (S) | Complex (S) | AVG (G) | Simple (G) | Normal (G) | Complex (G) |
> |:-------------------:|:----:|:-------:|:----------:|:----------:|:-----------:|:-------:|:----------:|:----------:|:-----------:|
> | InternVL3.5-8B       | 36.6 | 39.9    | 43.0       | 38.7       | 38.0        | 33.2    | 34.2       | 33.5       | 32.0        |
> | Qwen2.5-VL-7B        | **38.5** | **42.0** | 44.2       | 40.5       | 41.2        | **34.9** | 33.3       | 34.1       | 37.4        |
> | LLaVA-OneVision-7B   | 36.3 | 39.1    | 42.3       | 39.3       | 35.6        | 33.4    | 32.5       | 34.3       | 33.4        |
>
> > **Simple** prompt (Subunit-level)
>         > Describe the sign performed in the video from the following options: {question_choices}. Respond only with the selected option: A, B, C, or D.
>
> > **Normal** prompt (Subunit-level)
>         > In the middle of the input video, a person is performing American Sign Language (ASL). Each of the following options describes how a sign is performed, possibly using fingerspelled letters from American Fingerspelled Alphabet to indicate handshapes (e.g., A hand, F hand). Please select the option that best corresponds to the sign being performed: {question_choices}. Respond only with the selected option: A, B, C, or D.
>
> > **Complex** prompt (Subunit-level)
>         > You are tasked with analyzing a video segment where an individual is using American Sign Language (ASL). Your primary goal is to meticulously describe the production of a specific sign. To ensure a comprehensive and accurate analysis, you must consider all five fundamental parameters of ASL: handshape, palm orientation, location, movement, and non-manual markers. For clarity, (*detailed requirements for handshape, palm orientation, location, and movement*) ... If there are repeated movements, specify the number of repetitions. Please select the option that best corresponds to the sign being performed: {question_choices}. Respond only with the selected option: A, B, C, or D.
>
> 2. **Evaluating the domain-specific MLLMs.** As for fine-tuning MLLMs in the sign language domain, it is more appropriate to evaluate their effectiveness on downstream tasks, such as sign language recognition and translation. The proposed LexSign dataset and the HALI method offer *a valuable tool for evaluating the models’ ability to describe fine-grained sign language actions*. Specifically, they enable assessment of the quality and completeness of generated descriptions in ZSSLR experiments, providing a complementary perspective on the models’ linguistic understanding.
>
> Overall, the proposed LexSign-Bench provides a benchmark for evaluating the sign language understanding ability of general-domain MLLMs using fixed prompts, while domain-specific MLLMs can be assessed by the quality of their generated sign language captions, which is quantitatively evaluated by ZSSLR experiments.

---

> ### Author Response · Authors · 2025-11-24
> **Official Comment by Authors (3/4)**
>
> > [W3] about model selection and hyper-parameter tuning
>
> We are grateful for your valuable feedback. For the choices of evaluated MLLMs on LexSign-Bench, we select three advanced closed-source MLLMs and three open-source 7B/8B MLLMs. Although our current evaluation includes only six MLLMs, they are representative of the state of the art and thus provide reasonable and reliable baselines for LexSign-Bench. We adopt CoSign-1s as the default backbone, considering the robustness of skeleton-based models. We adopt BERT as the default textual encoder. We provide more details about the visual encoder, text encoder, and other model choices in the Method section and Appendix A.1.
>
> Regarding hyperparameter tuning for the proposed method, we provide ablation results on loss weights, batch size, and the number of selected subunit-level features. Our analysis shows that reducing the temperature in contrastive learning from 0.03 to 0.01 leads to a notable performance gain, improving the validation Top-1 accuracy from 35.3% to 37.1% on LexSign-ASL1000. We sincerely thank reviewer WGdq's suggestion and will update all experimental results in the next version. A more detailed hyperparameter ablation study can be found in Appendix A.11.
>
> *Ablation(%) on loss weights in the ZSSLR experiments on LexSign-ASL1000.*
>
> | $w_{\text{FG},s}$ | $w_{\text{FG},u}$ | $w_{\text{HC}}$ | Val Top-1 | Val Top-5 | Test Top-1 | Test Top-5 |
> |:---:|:---:|:---:|:---:|:---:|:---:|:---:|
> | 0.5 | 1.0 | 1.0 | 34.3 | 62.0 | 26.2 | 52.7 |
> | 2.0 | 1.0 | 1.0 | 35.1 | 63.2 | 27.1 | 53.4 |
> | 1.0 | 0.5 | 1.0 | 33.8 | 62.6 | 26.7 | 53.2 |
> | 1.0 | 2.0 | 1.0 | 35.0 | 62.6 | 26.8 | 53.1 |
> | 1.0 | 1.0 | 1.0 | **35.3** | **63.8** | 27.6 | 53.1 |
> | 1.0 | 1.0 | 0.5 | 35.0 | 63.3 | 27.2 | 53.0 |
> | 1.0 | 1.0 | 2.0 | 35.0 | 61.8 | 27.3 | 52.9 |
>
> *Ablation(%) on batch size and temperature on LexSign-ASL1000.*
>
> | $B$ | $\tau$ | Training Loss | Val Top-1 | Val Top-5 | Test Top-1 | Test Top-5 |
> |:---:|:------:|:-------------:|:---------:|:---------:|:----------:|:----------:|
> | 12  | 0.01   | $\mathcal{L}_{\text{Global}}$ | 30.9      | 58.1    | 23.1   | 48.6    |
> | 12  | 0.01   | $\mathcal{L}_{\text{HALI}}$   | **37.1**  | **64.3**  | 28.5   | 54.3  |
> | 12  | 0.1    | $\mathcal{L}_{\text{Global}}$ | 17.7      | 42.2  | 13.8    | 33.9  |
> | 12  | 0.1    | $\mathcal{L}_{\text{HALI}}$   | 30.7      | 57.9  | 24.2     | 48.7   |
> | 12  | 0.03   | $\mathcal{L}_{\text{Global}}$ | 26.5   | 54.2 | 21.0    | 44.2  |
> | 12  | 0.03   | $\mathcal{L}_{\text{HALI}}$   | 35.3   | 63.8 | 27.6   | 53.1  |
> | 4   | 0.03   | $\mathcal{L}_{\text{Global}}$ | 24.6   | 53.8 | 19.8    | 44.1  |
> | 4   | 0.03   | $\mathcal{L}_{\text{HALI}}$   | 31.0   | 60.1 | 23.3    | 49.7   |
> | 32  | 0.03   | $\mathcal{L}_{\text{Global}}$ | 26.4   | 54.0 | 20.1   | 43.7 |
> | 32  | 0.03   | $\mathcal{L}_{\text{HALI}}$   | 34.7   | 63.1 | 27.5    | 53.1    |
>
> *Ablation(%) on the number of selected subunit-level features on LexSign-ASL1000.*
>
> | Training Loss | $M$ | Val Top-1 | Val Top-5 | Test Top-1 | Test Top-5 |
> |:-------------:|:---:|:---------:|:---------:|:----------:|:----------:|
> | $\mathcal{L}_{\text{HALI}}$ | 2  |   33.3  |  61.0   |   26.8    |  52.5    |
> | $\mathcal{L}_{\text{HALI}}$ | 3 | **35.3**  | **63.8**  | 27.6   | 53.1 |
> | $\mathcal{L}_{\text{HALI}}$ | 5  | 34.6   | 62.4      | 26.6       | 52.9   |
>
> > [Q1] about the techniques to obtain fine-grained representation
>
> Sign language conveys meaning through explicit sequential and simultaneous composition of sign language subunits [r1]. Therefore, an isolated sign can be temporally decomposed into a sequence of consecutive sub-actions, each of which can further be spatially decomposed into multiple subunits. We intend the snippet-level representation to align with sub-actions, the subunit-level representation with actions of a subunit, and the fragment-level representation to correspond to either a sub-action or a subunit.
>
> We designed a rule-based method to extract snippet-level, subunit-level, and fragment-level representations based on the above motivation. In summary, for snippet-level representation, we perform pyramidal aggregation on the 1/2- and 1/4-scale frame features, producing a snippet-level representation of length 6 (=2+4) for each isolated sign. For subunit-level representation, we aggregate the left-hand, right-hand, and body features captured at these scales to construct the subunit-level representation, resulting in a subunit-level representation of length 18 (=3*6) for each isolated sign. For fragment-level representation, we primarily use the naturally occurring punctuation marks in the lexical descriptions, such as commas and periods, to segment each lexical description into fragments. The procedures for producing these fine-grained representations have been incorporated into the implementation details section.
>
> [r1] Sandler W, Lillo-Martin D C. Sign language and linguistic universals[M]. Cambridge University Press, 2006.

---

> ### Author Response · Authors · 2025-11-24
> **Official Comment by Authors (4/4)**
>
> > [Q2] about method comparison
>
> As the previous work [r1] does not release their dataset split under the GZSL setting, their reported results under the GZSL-S/U/H settings on ASL-Text are not directly comparable with our re-implemented or proposed results in Table 1. In contrast, all experimental results under the ZSL setting in Table 1, as well as all results in Tables 2 and 3, are fair comparisons, as they use consistent training and testing splits.
>
> [r1]. Bilge Y C, Cinbis R G, Ikizler-Cinbis N. Towards zero-shot sign language recognition[J]. IEEE transactions on pattern analysis and machine intelligence, 2022, 45(1): 1217-1232.
>
> > [Q3] about quality evaluation for description generation
>
> We sincerely thank the valuable suggestion of reviewer WGdq. We conducted additional experiments to better evaluate the quality of the automatically generated descriptions (AGD). On the LexSign-ASL1000 dataset, we start with dictionary-sourced descriptions for all 1000 training classes and progressively replace varying numbers of classes (denoted as *#AGD classes* in the table) with automatically generated descriptions, and report top-1 ZSSLR accuracy on (1) all dictionary-sourced descriptions, (2) all automatically generated descriptions below. Higher performance indicates higher quality of the training set descriptions.
>
> | #AGD classes | 0    | 100  | 200  | 300  | 400  | 500  | 600  | 700  | 800  | 900  | 1000 |
> |:------------:|:----:|:----:|:----:|:----:|:----:|:----:|:----:|:----:|:----:|:----:|:----:|
> | dictionary-based | **35.3** | 33.3 | 31.5 | 28.4 | 26.2 | 23.7 | 22.2 | 19.3 | 17.4 | 12.8 | 9.7  |
> | automatically generated | **9.7**  |  8.9 | 9.2  |  7.6 | 8.5  |  8.7 |  8.6 | 9.4  | 9.2  | 9.4  | 9.0  |
>
> It is observed that for the dictionary-based test set, increasing the proportion of automatically generated descriptions in the training set leads to lower performance. This indicates that the quality of the automatically generated descriptions still lags behind that of the dictionary sources, highlighting the challenges of fine-grained modeling.
>
> > [Q4] about the performance of general purpose MLLMs versus training ad-hoc sign models
>
> To provide a more thorough evaluation of MLLMs’ performance on LexSign-Bench, we conducted both a user study and a comparative analysis with conventional deep learning models. In the user study, we enlisted 10 signers and 10 non-signers, each of whom completed a 32-question survey spanning two difficulty levels and four iconicity types. This enabled us to collect human performance data from both groups (signers and non-signers) for direct comparison. For the conventional baselines, we used our proposed ZSSLR model for subunit-level evaluation and the ISLR model for gloss-level evaluation. Both models were trained on the LexSign dataset **with no benchmark data overlap**. The results are summarized below, with additional details provided in the main paper and Appendix A.9.
>
> | MLLMs | AVG (S) | AVG (G) | &nbsp;&nbsp; | Settings | AVG (S) | AVG (G) |
> | :--- | :---: | :---: | :---: | :--- | :---: | :---: |
> | *Closed-source MLLMs* | | | | *Human* | | |
> | GPT-5 | 62.6 | 65.7 | | Signer | 85.0 | 85.6 |
> | Gemini 2.5 Pro | 58.0 | 60.0 | | Non-signer | 78.1 | 62.5 |
> | Qwen-VL-Max | 48.3 | 39.9 | | *Conventional Models* | | |
> | *Open-source MLLMs* | | | | ZSSLR Model (full training set) | 51.3 | - |
> | InternVL3.5-8B | 38.7 | 34.3 | | ISLR Model (full training set) | - | 72.1 |
> | Qwen2.5-VL-7B | 40.5 | 34.4 | | ISLR Model (2/3 training set) | - | 71.0 |
> | LLaVA-OneVision-7B | 39.3 | 35.5 | | ISLR Model (1/3 training set) | - | 65.9 |
> | | | | | ISLR Model (1/6 training set) | - | 53.0 |
>
> In summary, GPT-5 demonstrates strong performance on gloss-level questions, achieving an accuracy of 65.7%, which is comparable to the ISLR model trained on one-third of the WLASL dataset (65.9%). Moreover, GPT-5 outperforms the ZSSLR model on subunit-level questions, indicating superior generalization capability. Notably, GPT-5 even exceeds the performance of non-signers on gloss-level tasks (65.7% vs. 62.5%), highlighting its impressive aptitude for sign language understanding. Collectively, these comparisons indicate that state-of-the-art MLLMs possess substantial sign language understanding ability (under four-choice questions). However, as illustrated in Tables 2 and 5, they still fall short of generating consistently reliable descriptions.

---

### Official Review · Reviewer_Cjx2 · 2025-10-30

**Soundness:** 2
**Presentation:** 2
**Contribution:** 2
**Rating:** 4
**Confidence:** 4

**Summary:**

This paper proposes to model sign language representations by introducing lexical descriptions. To this end, the authors extend the WLASL and DEVISIGN datasets with annotations that include lexical descriptions. They then learn sign language representations by aligning sign language videos with these descriptions. Additionally, the authors introduce LexSign-Bench, a benchmark designed to evaluate the sign language understanding capabilities of multimodal large language models (MLLMs).

**Strengths:**

**Strengths:**

- Although the authors do not clearly define what they mean by “lexical descriptions,” the attempt to exploit a new data modality to enrich sign-language features is a welcome effort.
- The proposed approach yields modest improvements on isolated sign-language recognition (isolated SLR).

**Weaknesses:**

**Weaknesses:**
- The paper never provides a rigorous definition of “lexical descriptions.” From the given examples they appear to be either an augmented gloss or a short action caption rather than a faithful translation of the sign video. The authors offer no justification for aligning videos to this coarse-grained surrogate instead of to the actual translation text, nor do they explain what benefit such a proxy signal could bring to end-to-end sign-language translation (SLT).
- The technical contribution is thin. The work focuses almost exclusively on data curation, while the algorithmic side adds little novelty. Video–text contrastive learning has become commonplace in SLT, and the proposed method scarcely departs from existing pipelines.
- I question the adequacy of off-the-shelf video-understanding models for sign language. Without targeted finetuning on sign-language data, these models are likely to produce only superficial or even misleading representations, undermining both the alignment phase and the subsequent evaluation on LexSign-Bench.

**Questions:**

- Could the authors provide a formal definition of “lexical descriptions” and explain why they choose this specific form of annotation—rather than the actual translation text—as the alignment target? How does this coarse-grained proxy benefit downstream tasks, particularly end-to-end sign language translation?
- While the introduction of lexical descriptions is interesting, the core algorithm appears to follow standard video–text contrastive learning. Could the authors clarify the novel technical contributions that distinguish this work from existing sign-language representation methods?
- The paper uses off-the-shelf video understanding models without sign-language-specific fine-tuning. Have the authors evaluated whether these models capture sign-language-specific visual-linguistic patterns? If not, how reliable are the learned representations and the subsequent benchmark results?

---

> ### Author Response · Authors · 2025-11-24
> **Official Comment by Authors (1/4)**
>
> We would like to express our sincere gratitude to the reviewers for their thorough assessment and valuable suggestions, which have significantly contributed to improving this work.
>
> > [W1 and Q1] about definition and benefit of lexical description
>
> Thank you for the insightful comments. We have revised the paper to prominently present the precise definition of the lexical description for better readability. The term lexical description refers to the comprehensive representation of sign language used by linguists to fully define and analyze a sign **as a distinct, meaningful unit of vocabulary (a lexeme)**, which contains detailed information about the sign’s location, handshape, movement, and orientation relative to other body parts [r1, r2]. In other words, a lexical description is intended to capture all essential components that constitute a sign, rather than serving merely as an augmented gloss or a brief action description. Different from general sign language transcription, lexical descriptions focus on the core linguistic properties of each sign (e.g, handshape, movement, location, and orientation) rather than producing a linear sequence of words or sentences. In this paper, we broaden the concept of lexical description to encompass **both traditional lexical definitions provided in sign language dictionaries and the automatically generated descriptive captions**.
>
> Recent works [r3, r4] have also attempted to use **automatically generated** sign language descriptions to enhance sign language translation (L158–159). **However, the performance gap between manually curated linguistic definitions and automatically generated descriptions remains insufficiently understood.** The proposed LexSign-Bench provides a valuable benchmark for evaluating MLLMs’ sign language understanding capabilities, while the HALI framework further enhances the use of lexical descriptions for downstream tasks such as sign language recognition and translation. This approach enables fine-grained representations, supports potential translation across dialects and popular slang, and facilitates detailed motion generation [r5].
>
> Reference
> - [r1] Battison R. Phonological deletion in American sign language[J]. Sign language studies, 1974, 5(1): 1-19.
> - [r2] Sandler W, Lillo-Martin D C. Sign language and linguistic universals[M]. Cambridge University Press, 2006.
> - [r3] Asasi S, Lakhal M I, Sincan O M, et al. Beyond Gloss: A Hand-Centric Framework for Gloss-Free Sign Language Translation[J]. arXiv preprint arXiv:2507.23575, 2025.
> - [r4] Kim J, Jeon H, Bae J, et al. Leveraging the power of mllms for gloss-free sign language translation[C]//Proceedings of the IEEE/CVF International Conference on Computer Vision. 2025: 21048-21058.
> - [r5] Bensabath L, Petrovich M, Varol G. Text-Driven 3D Hand Motion Generation from Sign Language Data[J]. arXiv preprint arXiv:2508.15902, 2025.

---

> ### Author Response · Authors · 2025-11-24
> **Official Comment by Authors (2/4)**
>
> > [W2 and Q2] about novelty and technical contribution
>
> We thank the reviewer for the insightful comment. While our work indeed places substantial emphasis on curating high-quality lexical descriptions, we would like to clarify that the technical contribution extends beyond data curation.
>
> - As mentioned above, some recent works [r1, r2] have also attempted to use **automatically generated** sign language descriptions to enhance sign language translation (L162–164). The constructed large-scale LexSign provides a resource that includes linguistic lexical definitions, generated descriptions, and a benchmark designed to **evaluate the sign language understanding capabilities of MLLMs**. Besides, the proposed LexSign dataset and HALI not only demonstrate the effectiveness of fine-grained lexical information in sign language recognition but also provide a means to **quantitatively evaluate the effectiveness and quality across various sources of lexical descriptions**.
>
>     Specifically, on the LexSign-ASL1000 dataset, we start with dictionary-sourced descriptions for all 1000 training classes and progressively replace varying numbers of training classes (denoted as *#AGD classes* in the table below) with automatically generated descriptions, and report ZSSLR top-1 accuracy on the test set of (1) all dictionary-sourced descriptions, (2) all automatically generated descriptions. Experimental results show that the quality of the automatically generated descriptions still lags behind that of the dictionary sources, highlighting the challenges of fine-grained modeling.
>
> | #AGD classes | 0    | 100  | 200  | 300  | 400  | 500  | 600  | 700  | 800  | 900  | 1000 |
> |:------------:|:----:|:----:|:----:|:----:|:----:|:----:|:----:|:----:|:----:|:----:|:----:|
> | dictionary-based | **35.3** | 33.3 | 31.5 | 28.4 | 26.2 | 23.7 | 22.2 | 19.3 | 17.4 | 12.8 | 9.7  |
> | automatically generated | **9.7**  |  8.9 | 9.2  |  7.6 | 8.5  |  8.7 |  8.6 | 9.4  | 9.2  | 9.4  | 9.0  |
>
> - We propose an Automated Generation Pipeline (AGP) that leverages large foundation models to generate high-quality lexical descriptions, as demonstrated in a quantitative result provided in Table 6 (and listed below) and a qualitative example provided in Fig.7. Unlike previous works that generate instance-wise video captions [r1, r2], the proposed framework **enhances cross-instance consistency while reducing intra-instance variability**, making it well-suited to provide lexical descriptions beyond those found in dictionaries.
>
> | #Instances | Qwen-VL-Max Top-1 | Qwen-VL-Max Top-5 | GPT-5 Top-1 | GPT-5 Top-5 |
> | :---: | :---: | :---: | :---: | :---: |
> | 1 w/o Summarizing | 2.1 | 8.5 | 4.7 | 16.7 |
> | 2 | 2.2 | 7.4 | 4.6 | 16.3 |
> | 3 | 2.5 | 9.2 | 5.9 | 18.4 |
> | 3 w/ Sampling | 2.3 | 9.3 | **6.7** | **22.1** |
>
> - The proposed HALI leverages **the hierarchical alignment between different levels of vision-language features in sign language (e.g., global hand-body posture and detailed finger movement)** to provide more precise linguistic grounding and demonstrate improved generalization on both existing and newly collected datasets. Unlike previous works [r3, r4] that rely on keypoint sequences or word embeddings, HALI exploits **the natural hierarchical structure within lexical descriptions**, making it more compatible with MLLMs and enhancing generalization. This approach not only achieves better performance than prior methods but also has the potential to support translation across dialects and popular slang.
>
> Reference
> - [r1] Kim J, Jeon H, Bae J, et al. Leveraging the power of mllms for gloss-free sign language translation[C]//Proceedings of the IEEE/CVF International Conference on Computer Vision. 2025: 21048-21058.
> - [r2] Bensabath L, Petrovich M, Varol G. Text-Driven 3D Hand Motion Generation from Sign Language Data[J]. arXiv preprint arXiv:2508.15902, 2025.
> - [r3]. Zuo R, Wei F, Mak B. Natural language-assisted sign language recognition[C]//Proceedings of the IEEE/CVF conference on computer vision and pattern recognition. 2023: 14890-14900.
> - [r4]. Wong R, Camgoz N C, Bowden R. Learnt contrastive concept embeddings for sign recognition[C]//Proceedings of the IEEE/CVF International Conference on Computer Vision. 2023: 1945-1954.

---

> ### Author Response · Authors · 2025-11-24
> **Official Comment by Authors (3/4)**
>
> > [W3 and Q3] about adequacy of off-the-shelf MLLMs for sign language understanding
>
> We demonstrate that current off-the-shelf MLLMs are already capable of capturing sign-language-specific visual–linguistic patterns through **two quantitative experiments** and **two qualitative examples**.
>
> **MLLMs can understand sign language.**
>
> - The quantitative result on LexSign-Bench validate the sign language understanding capability of MLLMs. It is important to clarify that **all descriptions used in LexSign-Bench are collected from authoritative dictionaries**, ensuring the benchmark’s reliability. We conducted a user study and compared MLLMs with conventional deep learning models on LexSign-Bench (see the table below and Tables 14, 15, and 16 for details). Our findings show that GPT-5, the strongest model evaluated, even surpasses non-signers on gloss-level questions and outperforms ZSSLR models on subunit-level questions, supporting the claim.
>
> | MLLMs | AVG (S) | AVG (G) | &nbsp;&nbsp; | Settings | AVG (S) | AVG (G) |
> | :--- | :---: | :---: | :---: | :--- | :---: | :---: |
> | *Closed-source MLLMs* | | | | *Human* | | |
> | GPT-5 | 62.6 | 65.7 | | Signer | 85.0 | 85.6 |
> | Gemini 2.5 Pro | 58.0 | 60.0 | | Non-signer | 78.1 | 62.5 |
> | Qwen-VL-Max | 48.3 | 39.9 | | *Conventional Models* | | |
> | *Open-source MLLMs* | | | | ZSSLR Model (full training set) | 51.3 | - |
> | InternVL3.5-8B | 38.7 | 34.3 | | ISLR Model (full training set) | - | 72.1 |
> | Qwen2.5-VL-7B | 40.5 | 34.4 | | ISLR Model (2/3 training set) | - | 71.0 |
> | LLaVA-OneVision-7B | 39.3 | 35.5 | | ISLR Model (1/3 training set) | - | 65.9 |
> | | | | | ISLR Model (1/6 training set) | - | 53.0 |
>
> - In Fig.6, we present the reasoning trajectory and final output of GPT-5 on a gloss-level question from LexSign-Bench. Below, we show GPT-5's reasoning process when answering the questions. The trace illustrates that the model not only **identifies fine-grained articulatory details** in the signing video ('thumb-up close to the chest' is recognized by GPT-5), but also leverages its knowledge to **understand how signs are produced** (GPT-5 knows how to sign “first”). These complementary abilities enable GPT-5 to infer the correct gloss.
>
> [The reasoning trajectory of GPT-5 answering a gloss-level question] *…… Reviewing frames 25 to 31, I observed a potential 'thumb-up close to the chest' gesture by the individual. Compared with the ASL sign for “first,” it usually has the thumb pointing upwards …… Thus, my answer is A.*
>
> **While not always perfect, MLLM can generate lexical descriptions for sign language that are partially correct.**
>
> - We trained ZSSLR models using the MLLM-generated descriptions and evaluated them on dictionary-sourced descriptions. These models achieve non-trivial performance (see the table below and Table 5 for details), following the same trend observed in the LexSign-Bench results.
>
> | Model | Val Top-1 | Val Top-5 | Test Top-1 | Test Top-5 |
> | :--- | :---: | :---: | :---: | :---: |
> | InternVL3.5-8B | 0.9 | 4.7 | 1.3 | 4.1 |
> | Qwen2.5-VL-7B | 1.5 | 6.8 | 1.1 | 4.6 |
> | Qwen-VL-Max | 2.9 | 10.2 | 2.1 | 8.5 |
> | GPT-5 | 6.1 | 20.8 | 4.7 | 16.7 |
>
> - In Fig.7, three descriptions generated by an MLLM (specifically, GPT-5) within our Automated Generation Pipeline (AGP), together with their corresponding videos, are demonstrated. Here, we provide a description generated by the MLLM, alongside a dictionary-based description of the same gloss *Watermelon* for comparison. As can be observed, GPT-5 produces outputs with impressive accuracy (bolded in the examples). It accurately attends to visual cues that are crucial for the semantic interpretation of sign language, while disregarding irrelevant details, although it makes some errors in recognizing key handshapes.
>
> [The description generated by MLLM] *With the right C hand, **palm facing left**, **contacting the back of the left S hand, palm facing down**, in front of the chest, **move the right hand in small clockwise circles across the stationary left hand.***
>
> [The description extracted from the dictionary] *Tap the index-finger side of the right W hand, **palm facing left**, against the chin with a double movement. Then, with a double movement, flick the middle finger of the right 8 hand, palm facing down, off **the back of the left S hand, palm facing down**, **bouncing the right hand up slightly each time.***
>
> Altogether, the qualitative inspection, benchmark evidence, and downstream ZSSLR results consistently indicate that MLLMs are capable of capturing sign-language-specific visual–linguistic patterns and can therefore be effectively applied to sign language understanding tasks.

---

> ### Author Response · Authors · 2025-11-24
> **Official Comment by Authors (4/4)**
>
> > [W3 and Q3] about finetuning MLLMs on sign-language data
>
> **Fine-tuning MLLMs with LexSign data can further enhance their lexical descriptions generation capability.** We finetune the MLLM using the LexSign dataset and observed that the resulting finetuned model produced substantially higher-quality descriptions. Specifically, we finetune Qwen2.5-VL using the video-description pairs of 700 glosses, which are present in LexSign-ASL1000 but absent from LexSign-ASL300, thereby ensuring that no data leakage occurs for the downstream task. We then apply AGP with the finetuned MLLM to generate descriptions for the training set of LexSign-ASL300. Based on these automatically generated descriptions, we trained a ZSSLR model and evaluated it using manually collected lexical descriptions from sign-language dictionaries. All downstream evaluations are carried out on LexSign-ASL300, with results summarized below.
>
> |                       | Val Top-1 | Val Top-5 | Test Top-1 | Test Top-5 |
> |-----------------------|-----------|-----------|------------|------------|
> | *Closed-source MLLMs* |           |           |            |            |
> | GPT-5                 | 9.7       | 29.0      | 6.7        | 22.1       |
> | Qwen-VL-Max           | 2.9       | 13.1      | 2.3        | 9.3        |
> | *Open-source MLLMs*   |           |           |            |            |
> | Qwen2.5-VL-7B         | 1.6       | 6.0       | 0.9        | 4.3        |
> | Qwen2.5-VL-7B (**Finetuned**) | 2.9 (+1.3)   | 10.2 (+4.2)      | 1.6        | 6.5        |
>
> It is observed that the ZSSLR model trained with descriptions generated by the finetuned Qwen2.5-VL achieves a substantial performance improvement over using descriptions generated by the original model. This confirms that LexSign is a valuable resource for finetuning MLLMs to improve their lexical descriptions generation capability and sign language understanding capability.

---

### Official Review · Reviewer_HiSm · 2025-10-31

**Soundness:** 3
**Presentation:** 3
**Contribution:** 2
**Rating:** 4
**Confidence:** 4

**Summary:**

The paper provides a dataset and benchmark for isolated sign classification using subunit information, including natural language descriptions of them (partially ground and partially pseudolabelled). It provides some experiments training sign language recognition models (which perform better using subunit information) and zero-shot with MLLMs, which show a little bit of sign language understanding.

**Strengths:**

The idea of natural language descriptions for signs is interesting, and the experiments with MLLMs are interesting, especially that they do better at the gloss questions than the subunit questions. Nice variety of baselines/experiments that look sound and reproducible, including prompts in the appendix etc. It's hard to imagine the ultimate solution for sign language understanding looking like HALI (all the hand-engineering) but it does improve the scores.

**Weaknesses:**

Overall I think the contribution is sound but low-moderate novelty and low-moderate significance. Related work misses some relevant papers from Kezar et al. https://arxiv.org/abs/2310.00196 https://arxiv.org/abs/2411.03568v1. I think I've seen at least one other work recently that used natural language descriptions of signs for MLLM sign language pretraining, but it was maybe within a reasonable timeframe before the ICLR deadline.

The paper could mention the possibility that the benchmark is contaminated wrt LLMs, because it's built on top of previously publicly available data.

I found myself having to refer back to earlier parts of the paper frequently, like the presentation could have been clearer, but maybe I'm just tired.

**Questions:**

I'm open to increasing my rating to 6 for the following:
* I'd like to see more about how this work differs from the Kezar papers which I linked above
* It would be interesting to see some qualitative analysis of the MLLM model outputs. I don't see them in Appendix A.2. I'm a bit suspicious of how high the scores in Table 10 are, though I guess it's multiple choice. Maybe I missed it, but how many multiple choices are there? 4?


There are some weird/incorrect/insensitive phrasings about sign language towards the beginning of the paper, but the rest of the paper looks fine. My rating is conditioned on these being fixed. Specifically:

> "Sign language is a well-defined visual language"
This isn't really what "visual language" means (https://en.wikipedia.org/wiki/Visual_language). Sign language also isn't "a language", it's a category of language (cf. spoken language) that includes many languages.

> "sign language... is largely unintelligible to hearing individuals"
"Unintelligible" has an unfortunate connotation that the thing can't be understood, rather than it isn't understood. I would just say "but it is not largely known by hearing individuals"

> " To bridge this communication gap, vision-based sign language understanding (SLU)"
It's a bit weird to phrase it this way when it's only half the communication gap.

> "enabling automatic recognition and translation of sign language from video input into textual or symbolic representations in a non-intrusive manner"
I wouldn't say this has been "enabled" yet; sign language models are still terrible. It aims to enable this

> "As a well-defined natural language"
Same as above, it's not "a" language.

Another misc comment: The acronym "ZSSLR" is introduced without explicitly introducing that it stands for zero-shot sign language recognition, ZSL, GZSL etc.

---

> ### Author Response · Authors · 2025-11-24
> **Official Comment by Authors (1/4)**
>
> We sincerely thank the reviewers for their time and thoughtful comments. Their insights have helped us improve the clarity and quality of our work.
>
> > [W1] About related work missing, low-moderate novelty, and low-moderate significance
>
> We appreciate Reviewer HiSm’s positive assessment of the soundness of our technical contribution and for highlighting the missing references. We acknowledge the concerns regarding novelty and significance, and we will use this opportunity to more clearly articulate the core distinctions between our proposed method and prior work.
>
> Kezar papers [r1, r2, r3] are pioneering efforts that leverage **linguistic information** to guide the recognition process, and our method benefits considerably from their insights. For example, we incorporate iconicity categories of glosses derived from the ASL-LEX annotations (L257–258). Collectively, these studies explore how symbolic linguistic structures can support sign language recognition. As highlighted by Reviewer HiSm, prior works [r4, r5] have also attempted to use **automatically generated** sign language descriptions to enhance sign language translation (L162–164). **However, the performance gap between manually curated linguistic definitions and automatically generated descriptions remains insufficiently understood.**
>
> To address this gap, we construct LexSign, a resource that includes linguistic definitions, generated descriptions, and a benchmark designed to **evaluate the sign language understanding capabilities of MLLMs**. Besides, the proposed HALI not only demonstrates the effectiveness of fine-grained lexical information in sign language recognition but also provides a means to **quantitatively evaluate the effectiveness and quality across various sources of lexical descriptions**. Notably, the comparison between Table 2 and Table 5 reveals that the descriptions generated by MLLMs remain far from satisfactory, highlighting the necessity of improving the quality of generated descriptions and exploring further enhancements.
>
> Regarding detailed comparisons with existing works, unlike Sem-Lex [r1], which constructs a large-scale curated video dataset with phoneme-level annotations and demonstrates the effectiveness of **explicit and disentangled** phonological features to improve sign language recognition, the proposed LexSign augments existing datasets to enable a more general comparison with state-of-the-art methods, **considering both manually curated linguistic definitions and automatically generated descriptions**. ASLKG [r2] leverages expert knowledge to build a knowledge graph combining ASL signs, English words, ASL phonemes, and ASL semantic features, employing neuro-symbolic methods for sign language understanding and evaluating its performance on ISLR. Different from this work that relies on a constructed knowledge graph, the proposed HALI **is more flexible with annotation sources and fully compatible with MLLMs**, making it better suited for scenarios not covered by a knowledge graph, such as dialects and popular slang.
>
> In summary, we adopt a more flexible approach that is compatible with MLLMs to generate and leverage lexical information. Note that our proposed descriptions can also be further combined with knowledge graph–based methods, as they can serve as entities in the knowledge graph. With respect to sign language pre-training, recent works [r4, r5] use sign language descriptions generated by MLLMs to enhance translation performance, without considering the quality of generated descriptions. The construction of LexSign provides a principled way to guide the selection of MLLMs and to evaluate both key visual-language alignment designs (e.g., HALI) and the quality of generated descriptions through ZSSLR results. We have added the discussion of these works in Section 2.1.
>
> Reference
> - [r1] Kezar L, Thomason J, Caselli N, et al. The sem-lex benchmark: Modeling asl signs and their phonemes[C]//Proceedings of the 25th International ACM SIGACCESS Conference on Computers and Accessibility. 2023: 1-10.
> - [r2] Kezar L, Munikote N, Zeng Z, et al. The American Sign Language Knowledge Graph: Infusing ASL Models with Linguistic Knowledge[C]//Findings of the Association for Computational Linguistics: NAACL 2025. 2025: 7017-7029.
> - [r3] Kezar L, Thomason J, Sehyr Z. Improving sign recognition with phonology[C]//Proceedings of the 17th Conference of the European Chapter of the Association for Computational Linguistics. 2023: 2732-2737.
> - [r4] Asasi S, Lakhal M I, Sincan O M, et al. Beyond Gloss: A Hand-Centric Framework for Gloss-Free Sign Language Translation[J]. arXiv preprint arXiv:2507.23575, 2025.
> - [r5] Kim J, Jeon H, Bae J, et al. Leveraging the power of mllms for gloss-free sign language translation[C]//Proceedings of the IEEE/CVF International Conference on Computer Vision. 2025: 21048-21058.

---

> ### Author Response · Authors · 2025-11-24
> **Official Comment by Authors (2/4)**
>
> > [W2] about contamination in LLMs
>
> We greatly appreciate this insightful comment. To further validate LexSign-Bench and exclude potential data leakage effects, we further evaluates MLLMs on three additional tests: (1) a blind test, where only the text is provided without the corresponding video; (2) a temporal perturbation test, in which the video sequence is randomly manipulated by dropping or repeating a few frames; and (3) a spatial perturbation test, where the signer and background are replaced using Wan2.2-Animate-7B. We present the result below and in Table 13, and provide some examples of the perturbed video instances in Fig.6.
>
> | | Blind Test |  |  | Original |  |  | Temp. Pert. |  |  | Spat. Pert. |  |  |
> | :--- | :---: | :---: | :---: | :---: | :---: | :---: | :---: | :---: | :---: | :---: | :---: | :---: |
> | MLLM | AVG | AVG (S) | AVG (G) | AVG | AVG (S) | AVG (G) | AVG | AVG (S) | AVG (G) | AVG | AVG (S) | AVG (G) |
> | ***Closed-source*** | | | | | | | | | | | | |
> | GPT-5 | 25.2 | 26.7 | 23.7 | 65.0 | 62.6 | 67.4 | 57.0 | 54.3 | 59.7 | 58.0 | 55.7 | 60.3 |
> | ***Open-source*** | | | | | | | | | | | | |
> | InternVL3.5-8B | 27.0 | 27.4 | 26.6 | 36.1 | 38.7 | 33.5 | 36.2 | 39.3 | 33.0 | 34.0 | 38.2 | 29.8 |
> | Qwen2.5-VL-7B | 27.6 | 27.6 | 27.7 | 37.3 | 40.5 | 34.1 | 35.4 | 39.3 | 31.5 | 33.8 | 37.7 | 30.0 |
> | LLaVA-OneVision-7B | 24.9 | 24.5 | 25.2 | 36.8 | 39.3 | 34.3 | 37.2 | 39.9 | 34.6 | 34.7 | 37.8 | 31.6 |
>
> As shown in the table, blind-test results remain close to the 25% random-chance accuracy **(each question has 4 choices)**, indicating that our benchmark is unbiased at the text-level. As for the video side, introducing temporal or spatial perturbations leads to a slight performance drop, which we attribute to the injected noise by temporal augmentation and diffusion models. However, the overall trend remains largely consistent, suggesting that current MLLMs do not benefit from prior exposure to the underlying datasets. GPT-5 maintains a clear advantage over all open-source MLLMs.
>
> > [W3] about paper readability and structure
>
> We have revised the specific issues you mentioned and have made every effort to revise other expressions throughout the manuscript. The primary modifications are summarized as follows.
>
> 1. Abbreviations and symbols have been made consistent throughout the paper, defined upon first appearance, and annotated in the headers of tables or captions of figures where they are used.
>
> 2. Key concepts, such as the definitions of lexical descriptions and fine-grained representations, are placed in prominent locations to help readers locate and understand them quickly.
>
> 3. Extensive experimental details have been updated in the Implementation Details section of the main text and in Appendix A.1, where we have made every effort to ensure the full reproducibility of our experiments and address the reviewers' concerns regarding the experimental setup.
>
> 4. The figure depicting the proposed Manual Curation Pipeline (MCP) and Automated Generation Pipeline (AGP) has been relocated to the main paper to improve clarity. We believe that the figure makes it easier to grasp AGP and MCP.

---

> ### Author Response · Authors · 2025-11-24
> **Official Comment by Authors (3/4)**
>
> > [Q1] about qualitative analysis of the MLLM model outputs
>
> We have added to Appendix A.3: (1) a lexical description produced by the Automated Generation Pipeline (AGP) for the gloss *Watermelon*, and (2) GPT-5’s reasoning and answer for a gloss-level question from LexSign-Bench where the correct answer is A (*First*).
>
> (1) Three descriptions generated by an MLLM (specifically, GPT-5) within our Automated Generation Pipeline (AGP), together with their corresponding videos, are demonstrated in Fig.7. Here, we provide a description generated by the MLLM, alongside a dictionary-based description of the same gloss *Watermelon* for comparison. As can be observed, GPT-5 produces outputs with impressive accuracy (bolded in the examples). It accurately attends to visual cues that are crucial for the semantic interpretation of sign language, while disregarding irrelevant details, although it makes errors in recognizing key handshapes.
>
> [The description generated by MLLM] *With the right C hand, **palm facing left**, **contacting the back of the left S hand, palm facing down**, in front of the chest, **move the right hand in small clockwise circles across the stationary left hand.***
>
> [The description extracted from the dictionary] *Tap the index-finger side of the right W hand, **palm facing left**, against the chin with a double movement. Then, with a double movement, flick the middle finger of the right 8 hand, palm facing down, off **the back of the left S hand, palm facing down**, **bouncing the right hand up slightly each time.***
>
> (2) We present the reasoning trajectory and final output of GPT-5 on a gloss-level question from LexSign-Bench in Fig.6. Below, we show GPT-5's reasoning process when answering the questions. The trace illustrates that the model not only **identifies fine-grained articulatory details** in the signing video ('thumb-up close to the chest' is recognized by GPT-5), but also leverages its knowledge to **understand how signs are produced** (GPT-5 knows how to sign “first”). These complementary abilities enable GPT-5 to infer the correct gloss.
>
> [The reasoning trajectory of GPT-5 answering a gloss-level question] *…… Reviewing frames 25 to 31, I observed a potential 'thumb-up close to the chest' gesture by the individual. Compared with the ASL sign for “first,” it usually has the thumb pointing upwards …… Thus, my answer is A.*
>
> > [Q1] about quantitative analysis of the LexSign-Bench result
>
> Yes, LexSign-Bench contains 7,296 multiple-choice questions, with four choices per question.
>
> To provide a more thorough evaluation of MLLMs’ performance on LexSign-Bench, we conducted both a user study and a comparative analysis with conventional deep learning models. In the user study, we enlisted 10 signers and 10 non-signers, each of whom completed a 32-question survey spanning two difficulty levels and four iconicity types. This enabled us to collect human performance data from both groups (signers and non-signers) for direct comparison. For the conventional baselines, we used our proposed ZSSLR model for subunit-level evaluation and the ISLR model for gloss-level evaluation. Both models were trained on the LexSign dataset **with no benchmark data overlap**. The results are summarized below, with additional details provided in the main paper and in Tables 14, 15, and 16.
>
> | MLLMs | AVG (S) | AVG (G) | &nbsp;&nbsp; | Settings | AVG (S) | AVG (G) |
> | :--- | :---: | :---: | :---: | :--- | :---: | :---: |
> | *Closed-source MLLMs* | | | | *Human* | | |
> | GPT-5 | 62.6 | 65.7 | | Signer | 85.0 | 85.6 |
> | Gemini 2.5 Pro | 58.0 | 60.0 | | Non-signer | 78.1 | 62.5 |
> | Qwen-VL-Max | 48.3 | 39.9 | | *Conventional Models* | | |
> | *Open-source MLLMs* | | | | ZSSLR Model (full training set) | 51.3 | - |
> | InternVL3.5-8B | 38.7 | 34.3 | | ISLR Model (full training set) | - | 72.1 |
> | Qwen2.5-VL-7B | 40.5 | 34.4 | | ISLR Model (2/3 training set) | - | 71.0 |
> | LLaVA-OneVision-7B | 39.3 | 35.5 | | ISLR Model (1/3 training set) | - | 65.9 |
> | | | | | ISLR Model (1/6 training set) | - | 53.0 |
>
> In summary, GPT-5 demonstrates strong performance on gloss-level questions, achieving an accuracy of 65.7%, which is comparable to the ISLR model trained on one-third of the WLASL dataset (65.9%). Moreover, GPT-5 outperforms the ZSSLR model on subunit-level questions, indicating superior generalization capability. Notably, GPT-5 even exceeds the performance of non-signers on gloss-level tasks (65.7% vs. 62.5%), highlighting its impressive aptitude for sign language understanding. Collectively, these comparisons indicate that state-of-the-art MLLMs possess substantial sign language understanding ability (under four-choice questions). However, as illustrated in Tables 2 and 5, they still fall short of generating consistently reliable descriptions.

---

> ### Author Response · Authors · 2025-11-24
> **Official Comment by Authors (4/4)**
>
> > [Q2] about weird/incorrect/insensitive phrasings about sign language
>
> We sincerely thank you for pointing out these issues, especially given the complexity of sign language and the importance of culturally appropriate terminology. We have revised the specific issues you highlighted and have carefully reviewed the manuscript to revise other expressions, ensuring that our language is respectful toward the Deaf and hard-of-hearing community.
>
> There is still a little question about "Sign language is a well-defined visual language". While Wikipedia does not classify sign language as a visual language, some literature defines visual language as "any kinds of non-textual but visible human communication medias, including ..., sign language, ...," [r1] and several earlier works also claim that "sign language is a visual language" [r2, r3]. To reduce potential ambiguity, we have revised the phrasing to use natural language and greatly appreciate your feedback on this change.
>
> - [r1]. Erwig M, Smeltzer K, Wang X. What is a visual language?[J]. Journal of Visual Languages & Computing, 2017, 38: 9-17.
> - [r2]. Bellugi U. Implications from a Visual Language[J]. Advances in cognition, education, and deafness, 1991: 11.
> - [r3]. Bowden R, Windridge D, Kadir T, et al. A linguistic feature vector for the visual interpretation of sign language[C]//European conference on computer vision. Berlin, Heidelberg: Springer Berlin Heidelberg, 2004: 390-401.

---

### Author Response · Authors · 2025-12-03
**Summary of All Responses (2/2)**

## References

[r1] Kim J, Jeon H, Bae J, et al. Leveraging the power of mllms for gloss-free sign language translation[C]//Proceedings of the IEEE/CVF International Conference on Computer Vision. 2025: 21048-21058.

[r2] Asasi S, Lakhal M I, Sincan O M, et al. Beyond Gloss: A Hand-Centric Framework for Gloss-Free Sign Language Translation[C]//Proceedings of the 36th British Machine Vision Conference. 2025: 626.

[r3] Kezar L, Thomason J, Caselli N, et al. The sem-lex benchmark: Modeling asl signs and their phonemes[C]//Proceedings of the 25th International ACM SIGACCESS Conference on Computers and Accessibility. 2023: 1-10.

[r4] Kezar L, Munikote N, Zeng Z, et al. The American Sign Language Knowledge Graph: Infusing ASL Models with Linguistic Knowledge[C]//Findings of the Association for Computational Linguistics: NAACL 2025. 2025: 7017-7029.

[r5] Kezar L, Thomason J, Sehyr Z. Improving sign recognition with phonology[C]//Proceedings of the 17th Conference of the European Chapter of the Association for Computational Linguistics. 2023: 2732-2737.

[r6] Kuang J, Wang H, Han C, et al. Zero-shot skeleton-based action recognition with dual visual-text alignment[J]. Pattern Recognition, 2025: 112342.

[r7] Zhou K, Zhang S, You Z, et al. Zero-shot skeleton-based action recognition with prototype-guided feature alignment[J]. in IEEE Transactions on Image Processing, vol. 34, pp. 4602-4617, 2025, doi: 10.1109/TIP.2025.3586487.

---

### Author Response · Authors · 2025-12-03
**Summary of All Responses (1/2)**

Dear reviewers and Area Chairs,

Thank you sincerely for reviewing and coordinating paper 7787. We greatly appreciate the reviewer's recognition of our approach as introducing **interesting ideas** of natural language descriptions for sign language (HiSm) with **significant real-world relevance** (hpXN), providing **sound and reproducible** experiments that **effectively support** our claims (HiSm, WGdq, hpXN), and proposing the dataset that can **greatly facilitate unified understanding and generation** of sign language (hpXN). We briefly recap the reviews as follows.
- **Reviewer HiSm** recognizes the interesting ideas and solid experiments in this work, while raising concerns about missing related works (Detail 1) and potential data contamination risks for MLLMs (Clarification 1), and suggests providing more qualitative analysis of the MLLM outputs (C1).
- **Reviewer Cjx2** raises concerns about the unclear definition of "lexical descriptions" (D2), the difference from other sign language representation methods (C2), and the adequacy of off-the-shelf MLLMs for sign language (C1).
- **Reviewer WGdq** finds the contributions sound and valuable, while expecting us to provide more details (D1) and include additional experimental results (C1).
- **Reviewer hpXN** acknowledges the importance of the proposed problem and dataset, and suggests addressing writing issues (D1), clarifying the motivation of the proposed method (C3), and expanding comparisons with more recent state-of-the-arts (C4).

---
We would like to emphasize four key clarifications below:

**C1. MLLMs have the capability to understand sign languages.**

We conduct thoughtful experiments to ensure evaluation results on the proposed benchmark are **not affected by prior exposure to public datasets (Appendix 8) or prompt engineering (A. 10)**. We also conduct a user study on both signers and non-signers (Section 5.2) to approximate the performance upper bound. Results (in Table 16) show that **closed-source MLLMs (e.g., GPT-5) achieve strong performance on gloss-level questions**, even outperforming non-signers' performance (65.7% vs. 62.5%).

As for lexical description generation, we also show that fine-tuning open-source MLLMs on domain-specific datasets **can yield a substantial performance gain but still lags behind closed-source MLLMs** (A.7).

[*A description generated by GPT-5*] With the right C hand, *palm facing left, contacting the back of the left S hand, palm facing down*, in front of the chest, *move the right hand in small clockwise circles across the stationary left hand*.

**C2. The technical contributions are not limited to the design of HALI.**

For clarity, we briefly restate the main contributions of this work:
- We construct **LexSign**, a high-quality dataset that includes both manually curated linguistic definitions and automatically generated descriptions. As [r1, r2] attempt to use automatically generated descriptions, this work provides **a valuable dataset to evaluate the quality across various sources of lexical descriptions**.
- We design **LexSign-bench**, a benchmark for comprehensively evaluating **the sign language understanding ability of MLLMs** as presented in C1.
- We propose **HALI**, a multi-granularity hierarchical alignment framework that captures the fine-grained structure of sign languages contained in lexical descriptions, and validate the effectiveness of lexical descriptions on ZSSLR/ISLR tasks.

**C3. This work attempts to explore the potential of lexical descriptions in SLR.**

In this work, we **broaden the concept** of "lexical description" to encompass both lexical definitions provided in sign language dictionaries and automatically generated descriptive captions. Different from [r3, r4, r5], the proposed HALI is **more flexible with annotation sources and fully compatible with MLLMs**, making it better suited for more challenging scenarios, such as **dialects and popular slang**.

**C4. HALI outperforms recent ZSL methods and can be combined with them for enhanced performance.**

We reproduce two recent ZSAR methods [r6, r7] and show that **our method consistently outperforms these baselines** (40.1% vs. 26.8%/17.0%). Incorporating HALI further brings significant performance gains (+11.2%/+2.7%, A.6) on these baselines, confirming the generalization ability of HALI.

---
We outline other detailed key improvements of our paper as follows.

**D1. Provide thoughtful discussion with related works [r3, r4, r5] (Section 2.1), implementation details (Section 5.1 and A.1), and selection strategies of MLLMs and the backbones (Section 5.2 and A.1).**

**D2. Incorporate definitions of key concepts, with inclusion of both lexical descriptions and fine-grained representations (Section 4.1).**

**D3. Improve the presentation of the paper, e.g., with clearer abbreviations and symbols.**

Thank you again for your time and effort in reviewing and coordinating our paper. We sincerely appreciate your consideration.

---

### Meta-Review · Area_Chair_ZZyD · 2026-01-02

**Summary:**

This paper initially recieved three 4 and one 6. Overall, the reviewers agree that the paper addresses an important and socially relevant problem in sign language understanding, with strengths in the construction of the LexSign dataset and benchmark, the exploration of lexical descriptions as a new linguistic supervision signal, and a comprehensive empirical evaluation involving conventional models and multimodal large language models. Several reviewers found the idea of leveraging fine-grained lexical structure promising and appreciated the breadth of experiments and reproducibility efforts.

Major weaknesses were consistently raised regarding limited novelty and technical depth: multiple reviewers consider the core method (HALI) as a relatively standard extension of existing video–text contrastive learning frameworks, with incremental algorithmic contribution compared to prior work.

Reviewers also questioned the unclear or initially imprecise definition of “lexical descriptions,” the reliance on off-the-shelf MLLMs without sign-specific adaptation, and the lack of strong theoretical motivation for the hierarchical losses. Additional concerns included evaluation and benchmark design issues (model selection protocol, prompt sensitivity, fairness of comparisons), writing clarity, and the risk that contributions rely heavily on dataset construction rather than methodological innovation, which limits impact for a top-tier vision venue

**Reviewer Concerns:**

On the positive side, the authors did make an effort to clarify what they mean by lexical descriptions and added more discussion and examples to explain this concept. They also expanded the experimental section, provided additional ablations, and addressed concerns about benchmark contamination and prompt sensitivity. These responses improve clarity and presentation compared to the original submission.

However, the central concern raised by multiple reviewers is still not fully resolved: what exactly constitutes a lexical description in practice and why it is the right representation to align with video? Even after reading the rebuttal, the definition remains broad and somewhat ambiguous, spanning dictionary-style linguistic definitions, generated captions, and coarse action descriptions. It is still unclear what level of granularity is expected, how consistent these descriptions are across sources, and how they differ in a principled way from existing representations such as glosses, phonological features, or action captions.

In addition, the rebuttal does not convincingly address the concern that achieving reliable subunit-level alignment or high-quality lexical descriptions would require substantial expert linguistic annotation. While automatic generation is emphasized, the results themselves suggest a large quality gap between generated and curated descriptions, raising questions about scalability and practical usefulness.

Finally, concerns about limited algorithmic novelty have been raised by several reviewers. While the hierarchical alignment framework is reasonable, it largely builds on standard video–text contrastive learning paradigms, and the rebuttal does not substantially strengthen the case that the technical contributions go beyond dataset construction and incremental architectural design.

**Reviewer Scores:**

The rebuttal makes an effort to clarify the notion of lexical descriptions, but the explanation remains insufficiently concrete, and it is still unclear what precisely constitutes a lexical description in practice and how it differs from existing representations. In addition, the core video–text alignment framework appears conceptually straightforward, and achieving reliable subunit-level alignment or linguistically faithful lexical descriptions likely requires substantial expert annotation that is not fully accounted for. As a result, the rebuttal does not resolve the reviewers’ key questions regarding conceptual clarity and technical contribution, and therefore does not provide sufficient grounds to revise the original evaluations.

---

### Decision · Program_Chairs · 2026-01-26

Reject